# Are climate change perceptions related with plastic policy support? Effects of climate change skepticism, guilt, and efficacy on the acceptance of the plastic tax

Miri Kim[1], Seoyong Kim[2]*, Sehyeok Jeon[3]

**1** Research Institute for Future Safety Policy, Ajou University, Suwon, Republic of Korea, **2** Department of Public Administration, Ajou University, Suwon, Republic of Korea, **3** Research Institute for Future Safety Policy, Ajou University, Suwon, Republic of Korea

* seoyongkim@ajou.ac.kr

## Abstract

Achieving a circular economy requires solutions to plastic pollution problems. Plastic waste poses significant threats to both human and biological systems globally. Plastics are closely related to climate change because their production is based on fossil fuels. Plastic taxation is one approach to reducing plastic use. This study aimed to analyze how the climate change perception is related to the preference for plastic tax. A model was developed to examine the impacts of 11 variables in value, risk perception, and planned behavior acting as predictors on the support for plastic tax as the predicted variable. In particular, we focused on both direct and indirect association of the three variables in planned behavior, that is, climate *skepticism, guilt, and efficacy about climate change crisis*, on support for the plastic tax. The results showed that among value variables, environmentalism, altruism, and egalitarianism had a significantly positive relationship with the willingness to pay the plastic tax and materialism had a significant negative relationship. Regarding risk perception, perceived risk, knowledge, and trust had a significantly positive association with the willingness to pay the tax. Regarding planned behavior, skepticism, guilt, and efficacy had a significant association with willingness-to-pay the plastic tax, with climate change skepticism having a negative effect and guilt and efficacy having positive relationship. Regarding the moderation effect, skepticism had an interaction effect on materialism and emotion; guilt is associated with the impact of environmentalism, altruism, egalitarianism, perceived risk on the support for plastic tax; and efficacy is associated with the association of altruism and perceived risk with preference for plastic tax. These results suggest that climate change risk perception is associated with plastic reduction behaviors.

**Data availability statement:** The data that support the findings of this study are feely available from https://doi.org/10.6084/m9.figshare.28735508.v2.

**Funding:** This work was supported by the Ministry of Education of the Republic of Korea and the National Research Foundation of Korea (NRF-2021S1A5C2A02087244). The Human Resources Development Project for HLW Management hosted by KORAD and MOTIE.

**Competing interests:** The authors have declared that no competing interests exist.

## Introduction

Plastic waste is occupying every corner of the planet. To analyze the factors that determine support for taxes to reduce plastic use, this study used climate change-related factors (e.g., skepticism, guilt, and efficacy) as key predictors, specifically evaluating their moderating effect on tax acceptance.

This study is necessary for three reasons. First, climate change and plastic waste issues are closely interconnected. Plastics are made from fossil fuels, and increased production results in increased greenhouse gases. Global plastics production reached 390.7 million metric tons in 2021, a 4% annual increase [1]. According to U.S. Beyond Plastics (October 2021), plastics will contribute more to climate change than coal-fired power plants by 2030 [2]. At least 14 million tons of plastic accumulate in oceans annually, with microplastics destroying marine ecosystems that help alleviate climate change effects [3].

Plastic pollution has reached unprecedented levels, with approximately 380 million tons produced annually [4], of which only 9% is recycled while 79% accumulates in landfills or natural environments. Marine ecosystems are particularly affected, with 11 million metric tons entering oceans annually [5], projected to triple by 2040 without intervention [6]. The plastic industry accounts for approximately 4–8% of annual global oil consumption [7], with greenhouse gas emissions potentially reaching 2.8 billion tons annually by 2050 [8].

Second, plastic tax introduction is garnering growing worldwide support. The European Union imposed a 0.8 euros per kilogram tax on non-recyclable plastic waste from January 2021 (Decision (EU) 2019/665). Italy became the first EU country to announce plastic tax at €1 per kilogram in 2020, targeting companies and consumers [9]. Indonesia also introduced plastic tax to address environmental damage. While US implementation faces challenges due to cost concerns for companies and consumers, understanding support levels for plastic tax legislation remains important.

Third, research on plastic tax implementation and human behavior effects is limited. Recent studies demonstrate significant behavioral impacts: Rivers et al. [10] found plastic bag taxes reduced consumption by 53–80% in Canada, while Homonoff et al. [11] documented substantial changes from small taxes on single-use bags. Taylor et al. [12] observed that plastic bag taxes were more effective than equivalent subsidies for eco-friendly alternatives. However, as plastic tax is new and previously unlevied, implementation research remains scarce. Despite increasing climate change policy importance, consumer plastic tax support research is limited, with plastic research focusing mainly on recycling technologies and eco-friendly materials [13].

This study explores how three factors (risk perception paradigm, value, planned behavior) associate with plastic tax support.

The conceptual foundation linking climate change perceptions to plastic tax support rests on three integrated theoretical frameworks. First, Values-Beliefs-Norms (VBN) theory demonstrates that environmental values associate with policy support through awareness of consequences [14], with stronger environmental values correlating with greater climate policy support [15]. Second, Theory of Planned Behavior

(TPB) suggests attitudes, subjective norms, and perceived behavioral control influence behavioral intentions [16]. Climate change skepticism significantly moderates environmental policy support [17], while guilt motivates pro-environmental behaviors when individuals perceive personal climate responsibility [18]. Third, risk perception paradigm indicates that perceived climate risks drive mitigation policy support [19], with higher climate risk perception correlating with increased environmental regulation support [20].

The link between climate change perceptions and plastic tax support emerges from plastic production's connection to greenhouse gas emissions, with plastic lifecycle contributing approximately 4–8% of global emissions [21]. Citizens perceiving climate change as serious risk more likely support emission-targeting policies, including plastic taxation [22].

Environmental policy acceptance requires multi-paradigm integration because single theories systematically underpredict behavior, explaining only 19–39% of variance [23]. Integration is theoretically necessary because environmental policies activate multiple psychological systems simultaneously. Values determine what risks matter [24], risk perceptions shape policy attitudes [25], and behavioral intentions require perceived control and normative support [16]. Combined models explain 52–58% of environmental behavior variance compared to 27–39% for single theories [26].

Our study focuses on plastic tax rather than direct prohibitions because economic incentives are often more effective for behavioral change. Rivers et al. [10] explained that taxes are more cost-effective than mandatory bans and generate less social resistance. Additionally, taxation represents an integrative approach affecting both consumers and producers [27]. We examined both direct and indirect impacts of skepticism, guilt, and climate change efficacy on plastic tax preference, particularly focusing on moderating roles within TPB.

The moderation role of planned behavior factors is theoretically grounded in TPB extensions. Climate change skepticism functions as a cognitive filter moderating value-behavior relationships through motivated reasoning [24], with skeptical individuals selectively processing information to confirm existing beliefs [28]. Guilt's moderating role is supported by affective-cognitive interaction theories, where emotional states amplify cognitive processing [29], enhancing relationships between environmental concerns and behavioral intentions [23]. Self-efficacy's moderation reflects social cognitive theory predictions that efficacy beliefs determine whether environmental values translate into behavioral intentions [30], strengthening value-behavior relationships by increasing confidence in meaningful contributions [31].

## Literature review

### The state of plastic world and public policy

During the COVID-19 pandemic, there has been a rapid increase in the use of plastics, which are a major source of greenhouse gas emissions. A prolonged pandemic would have led to graver environmental issues. The use of plastic during that period skyrocketed to the point that a new phrase, "corona trash," was coined. For example, the Korea Environment Agency compared the amount of plastic trash produced in 2020, when the pandemic began, with that in February 2022, and found that it increased 2.14 times, with a recycling rate of 68.1%, which was rather low compared to the increasing amount of plastic used.

In addition, the emergence of microplastics has increased plastic pollution. According to United Nations Environment Program [32], in the early 2000s, the amount of plastic waste produced rose more in a single decade than it had in the previous 40 years. Global production of primary plastic is expected to reach 1,100 million tons by 2050. The level of greenhouse gas emissions related to the production, use, and disposal of conventional fossil fuel-based plastics is estimated to account for 19% of the global carbon budget by 2040.

Plastic waste is now transforming into microplastics, which threaten natural and human ecosystems. A 2019 study by a team of researchers at Newcastle University in Australia, supported by the World Wide Fund for Nature [33], found that the average weekly per capita intake of microplastics is 5 grams. This means that microplastics are found in tap water, bottled water, streams, lakes, rivers, and oceans, in addition to the living environment and seafood worldwide.

Thus, more stringent policy measures are needed to regulate plastics. Accordingly, an international convention to regulate plastics is expected to be introduced in 2024, leading to stricter steps taken by major countries to regulate plastics. Many countries have increasingly regulated plastics, including the EU that imposed a plastic tax and phased out single-use plastics starting in 2021. It has implemented the Ecodesign Directive for almost all products, and is working on a legislation in which 0.8 euros per kilogram of plastic waste from domestic packaging will be paid to the EU, with 25% recycled content being required in plastic bottles by 2025 and 30% or more by 2030. The United States has banned single-use products such as plastic bags at the state level, China has banned the nationwide production and sale of single-use plastic products since 2021, and Japan is promoting the use of bioplastics with a goal of achieving 100% recycling by 2035. The United Kingdom has imposed a recycled plastic tax of about 300 won per kilogram, depending on the percentage of recycled content in the plastic.

A plastic tax is levied on non-recyclable plastic waste, and is being promoted to curb plastic use and promote the development of recyclable plastics. Because plastic taxes are emerging as the second trade barrier after carbon border taxes, it is necessary to discuss plastic taxes imposed worldwide. Imposing a plastic tax would contribute to solving the plastic waste problem, but it would be a financial burden on businesses and individuals, which makes acceptability a key issue. In terms of economic logic, plastic taxes can have regressive effects, disproportionately impacting lower-income households. Rivers et al. [34] found that environmental taxes create a higher relative burden on lower-income households who spend a larger proportion of their income on consumption goods. The financial impact varies with price elasticity. Homonoff et al. [11] demonstrated that plastic bag taxes reduced consumption by 40%, but elasticities differ across plastic product categories and household income levels. Economic modeling by Alpizar et al. [35] indicates that plastic taxes averaging 15–20% of product price can reduce household plastic consumption by 35%, but may increase household expenses by 1–3% annually, with lower-income households experiencing the higher percentage increase.

While plastic taxes create economic incentives for reduced consumption [10], direct interventions may yield more immediate impacts. Poortinga et al. [36] demonstrated that a plastic bag charge combined with environmental messaging produced significant behavioral changes, reducing bag usage by over 90%. Similarly, Nielsen et al. [22] found that social norm interventions combined with behavioral nudges effectively reduced single-use plastic consumption independently of price mechanisms.

However, it acknowledges that implementing a plastic tax would increase prices for plastic products, potentially creating inequality issues and affecting critical sectors like healthcare. As Lee et al. [37] noted, medical instruments rely on plastics for material integrity and hygiene reasons [37]. During crisis periods like the COVID-19 pandemic, a plastic tax without exemptions could impede medical supply access, disproportionately affecting vulnerable populations.

In the other hand, it considers the theoretical distinction between supporting regulatory policy versus engaging in personal plastic reduction behaviors. As Stern [14] distinguishes, environmental citizenship behaviors (like supporting taxes) differ from private-sphere behaviors (like reducing personal plastic use). Future research should investigate whether climate change awareness predicts both policy support and direct plastic reduction behaviors.

### Three competing paradigms for explaining the attitude toward climate change

Recent studies have expanded understanding of psychological mechanisms in environmental behaviors. Zhang et al. [38] demonstrated how cultural values influence low-carbon behaviors through perceived value mediation, providing precedent for examining value-behavior relationships in environmental policy contexts. Liu et al. [39] established management practices as mediators in green organizational behavior, paralleling our planned behavior factors as mediators between values and policy support. Zhang et al. [40] showed nature connectedness mediating outdoor activities and well-being, contrasting our climate perception focus while reinforcing environmental connection importance. Dai et al. [41] explored identity as mediator in tourism contexts, supporting our multi-pathway approach to environmental policy acceptance.

However, very few models explain the public response to plastic tax. We assumed that plastic taxes are associated with attitude and behavior toward climate change because plastics are closely linked to fossil fuel production, which results in climate change. The dominant explanatory theories of climate change-related behavior can be divided into value, risk perception, and planned behavior theories.

First, value theory recognizes that values play a key role in shaping individual attitudes toward environmental issues. Values are the fundamental orientations of individuals that are associated with their attitudes and behaviors. For example, the Values-Beliefs-Norms (VBN) theory proposed by Stern [14] explains pro-environmental behavior based on the three concepts of values, beliefs, and norms, and suggests selfish values, altruistic values, and ecological values as antecedents. According to VBN theory, values can be defined as ideas that regulate and influence human behavior; specifically, values determine individual beliefs and attitudes and are ideologies and principles that individuals, groups, and societies consider important [42]. Plastic use is linked to human habits. Changing habits requires a fundamental shift in our value system. From this perspective, examining the relationship between human values and plastic tax is necessary.

Second, the risk perception paradigm has been widely studied in the context of risky objects, such as nuclear energy. The risk perception paradigm, also known as the psychometric paradigm, was first discussed by Fischhoff et al. [43]. The risk perception paradigm is characterized by assuming risk as a subjective concept, with risk criteria including social and psychological aspects. It emphasized the importance of perception structures in risk judgment, and scholars focused more on evaluating public consciousness [44–46]. Previous studies have also analyzed risk perception in the context of climate change and global warming [47–49]. This study considers risk perception variables, such as *perceived risk, knowledge, government trust, and emotion*, as independent variables to determine the acceptance of the plastic tax.

Third, this study examined the direct and indirect roles of skepticism, guilt, and climate change-related efficacy. All three variables were derived from the theory of planned behavior (TPB) developed by [16,43,50]. TPB, one of the widely used theories to study the relationship between intention and action, assumes that the most important factor in determining "action" is "intention" [51]. Ajzen [16] acknowledged that not all intentions led to action and that intentions were frequently abandoned, or some were modified to respond to the situation. After conducting a meta-analysis of studies related to the theory of reasoned action and TPB, Sutton [52] stated that the reviewed theoretical models could explain approximately 40–50% of intention on average; however, the explanatory power was decreased to 19–38% when the intention is followed-up with the intended behavior. TPB is the most popular model used to explain human behavior in the context of climate change [31,53,54], demonstrating that the negative or positive attitude, norm, and climate change-related efficacy are associated with the response to climate change.

We aim to further analyze how the three variables derived from TPB—that is, climate change skepticism, guilt, and efficacy—play a role in the relationship between value/risk perception and support for the plastic taxes. This analysis can address the question of whether climate change awareness can drive the acceptance of tax for plastic reduction policies.

## Values factors

**Environmentalism.** With the increasing environmental crisis, the importance of environmental values has been acknowledged, and interest in environmentalism as an alternative has increased with an amplification of the harmful effects of environmental crisis. Environmentalism encompasses all philosophies, ideologies, and social movements that strive to protect the environment and promote environmental welfare, mainly in terms of preserving or restoring the natural environment.

Environmentalism has a negative impact on the acceptance of risky objects (e.g., nuclear energy or climate change). Spence [55] found a negative relationship between climate change concerns and nuclear energy acceptance, with environmental concerns reducing support for nuclear energy. Park et al. [56] found that environmentalism was statistically significant in the acceptance of energy price support policies for low-income households. Mozumder [57] found that environmental consciousness has a significant effect on the support for renewable energy. In addition, individuals who

possessed environmentalist values supported behaviors and policies related to environmental protection [58]. The relationship between environmental values and plastic consumption has been well-established by Heidbreder et al. [27], who demonstrated that pro-environmental values significantly predict reduced plastic consumption and support for plastic reduction policies. Because plastic is one of the risky objects such as nuclear power, we propose the following hypotheses.

**Hypothesis 1**. Higher environmentalism is associated with higher acceptance of the plastic tax.

**Altruism.** The VBN theory proposed by Stern [14] explains eco-friendly behavior based on three concepts, and divides values into altruistic, selfish, and ecological values. Altruistic values are pro-social values that focus on others rather than oneself, recognizing that individuals are interrelated to others. Whereas, egoistic values focus on personal success or wealth accumulation, which means that all thoughts and interests are focused on the self and not on others [59,60]. Ecological value refers to the value of the environment or ecosystem, rather than that of the individual or others, and prioritizes the preservation of ecosystems over humans. Jang et al. [61] showed that altruistic value orientation influenced the intention to purchase eco-friendly food products; in particular, the influence of altruistic value was more than that of individualistic or ecological values. As altruism is more likely to lead to favorable attitudes toward climate change, it can lead to negative attitudes toward plastic taxes.

**Hypothesis 2.** The higher the level of altruism, the higher acceptance of the plastic tax.

**Egalitarianism.** Douglas et al. [62] consider cultural theory crucial for explaining risk perception and the risk perception paradigm. Cultural theory seeks to explain how people perceive the world and act, and the way an individual perceives the world is determined by the way they live [63]. Cultural values consist of egalitarianism, hierarchy, individualism, and fatalism, and influence the attitudes of individuals [62]. Among the four values, egalitarianism has a strong degree of collectivization, but aims for equality among group members, and the external regulation of individual lives is rather weak. For egalitarians, nature is easily damaged, so they are sensitive to new technologies and environmental issues that may change the state of nature or pose irreparable risks to future generations. In addition, they demonstrate a negative attitude about the exercise of power by experts or institutions. Kahan et al. [24] found that individuals with individualistic values tend to be skeptical about risks such as global warming, while those with egalitarian values tend to agree with the view that global warming is a risk. In other words, because egalitarians are more sensitive to risk issues, they are more likely to be willing to pay the plastic tax to address climate change.

**Hypothesis 3**. Higher egalitarianism is associated with a higher acceptance of the plastic tax.

**Materialism.** Inglehart [64] explains that the advent of industrialized democracies has led to a shift from materialism to post-materialism by satisfying the most basic values and needs of economic growth and consumption [65]. In other words, while the satisfaction of physiological needs for human survival and the need for safety are directly related to materialism, the values that are pursued after materialistic values are largely realized are dematerialistic values, which are directly related to self-actualization and the main features of the community [66]. According to Kim et al. [65], environmentalism is a value that is mainly discussed in developed countries, and environmentalism boils down to the value of satisfying higher-order needs. Consequently, dematerialization has a positive effect on environmentalism and environmental behavior [67].

Materialistic values represent an important antecedent of attitudes within the TPB model, as evidenced in research by Hurst et al. [68], who demonstrated through meta-analysis that materialism negatively correlates with environmental attitudes and behaviors. Similarly, Ku [69] found that materialistic values significantly predicted less environmentally friendly behaviors. As Kilbourne et al. [70] argue, materialistic values predispose individuals toward attitudes that are less supportive of environmental policies, which directly relates to the attitude component of TPB.

 

In this study, we assume that the lower the level of materialism, the higher the level of support for the plastic tax, which implies an increased awareness of the fragility of nature and can be viewed as a corresponding behavioral intention to protect the environment.

**Hypothesis 4.** The lower the materialism, the higher acceptance of the plastic tax.

### Risk perception paradigm

**Perceived risk.** Perceived risk, one of the most powerful variables in terms of acceptability in risk research, refers to the subjective evaluation and perception of risk. According to Slovic et al. [71], perceived risk reflects various social and psychological factors such as uncertainty and controllability of risk, and focuses on how risk is perceived. In particular, compared with perceived benefits, perceived risk plays an opposing role in acceptability. Visschers et al. [72] found that perceived benefits and perceived risks were opposite and significant for acceptance. For renewable energy, similar to nuclear energy, perceived benefits have a positive effect on the support for renewable energy, while perceived risks have a negative effect [73]. Kim et al. [74] found that the higher the perceived risk of climate change and perceived benefits of climate change mitigation, the higher the intensity of climate change awareness and response behavior. As this study aims to identify the factors determining the support for taxes to address climate change, it is assumed that a higher perceived risk will lead to a higher support for payment.

**Hypothesis 5**. Acceptance of the plastic tax increases with an increase in the perceived risk of climate change.

**Knowledge.** Studies related to nuclear power acceptance and knowledge have shown that knowledge variables are useful in explaining acceptance [75]. Knowledge is an important factor that relates to acceptance of renewable energy and energy preference structure. Park et al. [73] found that knowledge influenced renewable energy acceptance through risk perception, and Langer [76] found that knowledge was a determinant of wind energy acceptance. O'Connor et al. [49] highlighted that more knowledge is associated with a higher intention to take voluntary action in response to climate change. Kim et al. [74] confirmed that knowledge affected the intention to act on climate change.

Recent research demonstrates that knowledge functions as a necessary but not sufficient condition within broader psychological processes [23]. As Kollmuss et al. [77] explain, knowledge contributes to pro-environmental behavior when mediated by personal values and situational constraints. Additionally, Gifford et al. [78] found that knowledge influences attitudes toward climate policies when combined with efficacy beliefs. Our findings support this integrated perspective rather than the simplistic knowledge-behavior model.

In our case, 'knowledge of climate change' is defined as objective and subjective understanding of climate science and related policies, measured through self-reported familiarity with climate change issues and government policies. This construct differs from acceptance of climate risk, which reflects the perceived severity and vulnerability associated with climate threats. As demonstrated by van der Linden [19], knowledge and risk perception represent distinct psychological constructs that influence climate policy support through different pathways. Knowledge primarily operates through cognitive processes, while risk acceptance involves affective judgments of potential harm.

In line with the previous studies, we believe that knowledge of climate change will have a significant effect on risk perception and the acceptance of the plastic tax. Thus, the following hypothesis is proposed.

**Hypothesis 6.** The higher the knowledge of climate change, the higher acceptance of the plastic tax.

**Trust.** Trust is a representative socio-cultural factor in risk research, and it plays an important role in improving individuals' risk perception, safety, and risk response effectiveness [79]. In risk research studies, trust can be defined as the level of confidence individuals have in organizations that provide information about risks and directly or indirectly

manage them [80]. Many studies have analyzed trust in the context of risk object, that is, nuclear acceptance [44,79,81]. Positive trust reduces risk perception and increases acceptance, which emphasizes the need for trust management. Visschers et al. [72] found that trust in the utility and government in nuclear power operations influenced the acceptance of nuclear power before and after the Fukushima nuclear accident. In this study, we believe that trust in the government is particularly important in the context of climate change policy, and reasons for distrust include a lack of confidence in the government's ability to manage nuclear energy, perceived unfairness, or a perception that the government does not share their values.

In this study, trust in government" construct specifically measures confidence in government capability and effectiveness to address climate change, rather than belief alignment. As Poortinga et al. [36] distinguish, this represents "competence trust" rather than "value similarity trust." Our items assess whether respondents believe the government's climate goals are achievable and whether the government is effectively addressing climate change.

Thus, the following hypothesis is proposed

**Hypothesis 7.** Increased trust in government regarding climate change will increase acceptance of the plastic tax.

**Emotions.** Emotions can be defined as good or bad feelings toward external stimuli. In particular, emotions are highlighted in relation to nuclear energy, emphasizing the inability of rational and reasonable judgment to operate [82]. Emotions are expressed as feelings, sentiments, and stigmas, and in nuclear research, they can be understood as the negative images that people have of nuclear power plants and nuclear containment centers [83]. Huijts et al. [84] confirmed that emotions were determinants of the attitude toward energy. Visschers et al. [72] confirmed that positive feelings in the affective dimension positively affected the acceptance of nuclear energy by lowering risk perception and increasing benefit perception. Kim et al. [74] confirmed that negative emotions had a significant impact on nuclear energy acceptance. Lim et al. [85] found that emotions had a negative impact on the perception of nuclear energy. In this study, we believe that negative emotions related to climate change will positively affect the support for and support future greenhouse gas reduction policies. Thus, the following hypothesis is proposed:

**Hypothesis 8.** The higher the negative emotions related to climate change, the higher acceptance of the plastic tax.

## Planned behavior factors

**Climate change skepticism.** Skeptical environmentalism, also known as environmental skepticism, refers to the belief that the environmental crisis claimed by scientists and environmental scientists is overestimated and that there are data errors [74]. Climate change skepticism is conceptually related to awareness of consequences (AC) from the VBN theory. As demonstrated by Poortinga et al. [36], climate skepticism fundamentally undermines the acknowledgment of consequences, serving as an inverse indicator of AC. Those with higher skepticism typically demonstrate lower awareness of environmental consequences.

Lomborg [86] found that skeptics with strong anti-environmentalist leanings believed that environmentalism was a threat to social and economic progress and civil liberties. Whitmarsh [87] found that skepticism was a structural problem, with the proportion of people believing in climate change being overestimated to more than double the value between 2003 and 2008. Climate change skepticism is a position that denies the seriousness of climate change, and is sometimes referred to as climate change denialism [73]. Albayrak et al. [88] analyzed the causal relationship between skepticism and environmental behavior and found that the weaker the skepticism, the stronger the pro-environmental purchasing behavior. Further, the stronger the skepticism, the lower the climate change awareness and intention to act [73]. Thus, the following hypothesis is proposed:

**Hypothesis 9.** Climate change skepticism moderates the relationship between climate change values, risk perception paradigm, and acceptance of the plastic tax.

**Guilt toward climate change.** Guilt is a response to personal responsibility for wrongdoing when some social standard has been violated [89]. Individuals who feel guilty want to undo their actions; guilt typically elicits a pro-social response, characterized by repairing the harm caused by their wrongdoing. Moreover, eco-guilt refers to a negative emotional state that occurs when people perceive that they have failed to meet personal or social standards for environment-friendly behavior. Guilt can often act as a motivating factor to make significant changes in behavior to become more eco-friendly. Ferguson et al. [90] demonstrated that climate change-related behavior was facilitated when groups, including individuals, perceived responsibility for climate change and felt guilty about it. Wang [91] explored the determinants of climate change behavioral intention and found that self-efficacy, guilt, attitude, and consideration of future consequences predicted behavioral intention to mitigate global warming. In this study, we examine the role of climate change guilt in moderating the relationship between value and risk perception factors and the acceptance of the plastic tax. Thus, the following hypothesis is proposed:

**Hypothesis 10.** Guilt moderates the relationship between climate change values and risk perception paradigms and acceptance of the plastic tax.

**Self-efficacy.** Self-efficacy is belief in one's abilities [92] and refers to the satisfaction felt by an individual regarding their role and actions related to their environment [65]. It is belief in one's ability to organize and sustain the activities necessary to achieve a specific goal. Self-efficacy is associated with cognitive functioning through its impact on self-satisfaction with personal development and the demands of selected goals, and is one of the best predictors of environmental behavior [93]. Lee et al. [94] found that self-efficacy had a significant positive effect on eco-friendly behavior. Kim et al. [65] also confirmed that self-efficacy had a positive influence on environmentalism. Tabernero et al. [95] found that self-efficacy triggered the intrinsic motivation to practice self-regulatory environmental behaviors such as recycling. Therefore, this study aims to investigate the effect of self-efficacy in climate change on the determinants of the support for the plastic tax.

**Hypothesis 11.** Self-efficacy moderates the relationship between climate change values, risk perception paradigm, and acceptance of the plastic tax.

**Planned behavior factor as a moderator.** In this study, we assumed that the three variables that comprise planned behavior play a moderating role in the relationship between value/risk perception and plastic tax. Logically, planned behavior is a powerful variable that can influence intentions and attitudes, as proven by prior studies [65,94,95]. Moreover, the moderating role of planned behavior factors such as attitude, norm, and efficacy has received increasing attention. Yang et al. [96] showed that the effects of an interaction between attention to efficacy messages and that to threat messages on behavioral intention vary among people with different attitudes, subjective norms, and perceived behavioral control. Norms influence a variety of climate change-related behaviors. Wang [91] showed that moral attitudes, self-efficacy, anticipated guilt, and consideration of future consequences predicted one's intentions to engage in behaviors to alleviate global warming. Moghavvemi [97] demonstrated that feelings of guilt about contributing to air pollution and an awareness of the advantages of using LED lights both activate individual personal norms and attitudes and influence the intention to buy LED lights. This study adopts climate change skepticism as the attitudinal variable that constitutes the planned behavior. According to Chen [98], the moderating effect of climate change skepticism on the positive relationship between pro-environmental self-identity and purchase intentions of sustainable labeled coffee was verified. Accordingly, the following hypothesis is proposed.

Hypothesis 11. Skepticism, guilt, and efficacy moderates the relationship between climate change values, risk perception paradigm, and the willingness to pay the plastic tax.

Based on motivated reasoning theory, climate change skepticism should weaken the positive relationships between pro-environmental values and plastic tax support, as skeptics discount environmental information [24]. Research demonstrates that skepticism attenuates the value-behavior link by reducing perceived policy necessity [17].

For guilt moderation, moral self-regulation theory suggests that guilt amplifies value-behavior consistency by heightening moral salience [99]. Studies show guilt strengthens the relationship between environmental concerns and policy support among high-guilt individuals [18].

Regarding efficacy, social cognitive theory indicates that self-efficacy enhances the translation of values into behavioral intentions when individuals believe their actions matter [30]. Research confirms efficacy moderates the environmental value-behavior relationship [100]. Based on those literatures, we propose specific hypotheses:

H9a: Climate change skepticism negatively moderates the relationship between environmentalism and plastic tax support

H10a: Guilt positively moderates the relationship between environmental values and plastic tax support

H11a: Self-efficacy positively moderates the relationship between perceived risk and plastic tax support.

## Method section

### Data collection and ethical statement

This study used nationally representative survey data for the analysis. A nationwide survey was conducted by a professional polling agency from May 30 to June 3, 2022, with 1,571 adults aged 19 and above. Participants were recruited through a professional polling agency using stratified random sampling based on region, gender, and age demographics matching Korean population statistics. This study employed a nationally representative cross-sectional survey design following established protocols [101]. A stratified random sample of 1,571 Korean adults (aged 19+) was recruited through a professional polling agency using proportional quota sampling by region, gender, and age. The web-based survey achieved a 95% confidence level with ±2.5% sampling error.

The web-based survey was administered from May 30 to June 3, 2022. Survey links were distributed via text messages and emails to eligible participants. The questionnaire employed block randomization to minimize order effects, with value measures, risk perception items, and planned behavior scales presented in randomized sequences across participants. Informed consent was obtained electronically before survey completion. The average completion time was approximately 15 minutes. This study was conducted in accordance with the Declaration of Helsinki principles and Korean ethical research standards. Ethical review exemption was granted under the Korean Bioethics and Safety Act (2005), Article 15 [2], which permits exemption for research meeting specific minimal risk criteria aligned with international standards.

Our study qualified for exemption under multiple established categories. First, the research involved minimal risk to participants, defined as risk no greater than that encountered in daily life activities. Survey questions addressed standard psychological constructs (attitudes, values, perceptions) commonly used in social science research without sensitive topics that could cause psychological distress. Second, our web-based methodology eliminated face-to-face contact, which under Korean regulations requires additional IRB oversight due to potential identification risks. The anonymous online platform ensured participant confidentiality throughout data collection, with no mechanism for linking responses to individual identities. Third, we collected no personally identifiable information as defined under Korean Personal Information Protection Act Article 23. Demographic variables (age, gender, education, income) were collected in categorical ranges without specific details that could enable individual identification.

Despite exemption status, we implemented comprehensive participant protection protocols. Electronic informed consent was obtained from all participants before survey access, including clear information about study purpose, voluntary participation nature, estimated completion time, data usage intentions, and unconditional right to withdraw without penalty. Participants received explicit assurance that responses would remain anonymous and confidential, with data stored on secure servers using encryption protocols. No personal contact information was collected or retained, ensuring complete anonymity throughout the research process.

We conducted thorough risk assessment demonstrating minimal potential for harm. Survey questions addressed commonly studied psychological constructs without probing sensitive personal experiences, traumatic events, or stigmatized behaviors. The environmental policy topic represents low-risk subject matter unlikely to cause emotional distress or social vulnerability. Data collection procedures minimized any residual risks through anonymous participation, voluntary response options for all questions, and clear withdrawal procedures accessible throughout survey completion.

Our exemption approach aligns with international research ethics frameworks, including U.S. Federal Policy for Protection of Human Subjects (45 CFR 46) categories for exempt research involving anonymous surveys on non-sensitive topics. Similar exemption criteria exist in European and other international contexts for minimal risk social science research. The Korean Bioethics and Safety Act criteria mirror international standards by distinguishing between research requiring full IRB review and minimal risk social science surveys warranting exemption with appropriate safeguards. Complete exemption documentation was maintained according to Korean regulatory requirements, including detailed protocol descriptions, risk assessments, and participant protection measures, with our research institution's ethics office confirming exemption appropriateness under applicable regulations.

The demographic characteristics of the respondents were as follows: 772 (49.1%) male and 799 (50.9%) female, with 255 (16.2%) respondents being aged between 19 and 29, 233 (14.8%) aged between 30 and 39, 294 (18.7%) aged between 40 and 49, 314 (20.0%) aged between 50 and 59, and 475 (30.2%) aged 60 and above. In terms of education, 792 (50.1%) had a high school diploma or less, and 779 (49.6%) had a college degree or higher. In terms of income, 1066 (67.9%) earned less than 6 million won, and 505 (32.1%) earned more than 6 million won.

**Measures and reliability**

The measurement items and reliability analysis are shown in Table 1, and the reliability test showed that the Cronbach's α coefficient ranged from 0.635 to 0.891, confirming that the reliability values were all met.

All of the questions were measured using a 5-point Likert scale except altruism, and the mean value was used as a variable.

Our scale items were carefully selected from established literature with strong psychometric properties. For climate change skepticism, we adopted items from Whitmarsh [87], who developed and validated a comprehensive skepticism scale. This approach aligns with McCright et al's [102] work in Global Environmental Change measuring public climate skepticism. Our guilt measurement follows Ferguson et al. [90] in Journal of Environmental Psychology, who demonstrated how collective guilt mediates climate change beliefs and mitigation behaviors. Harth et al. [89] similarly validated guilt measures predicting distinct environmental intentions. For efficacy, we utilized measurements from Bandura's [93] framework, following Lauren et al.'s [100] application in Journal of Environmental Psychology showing self-efficacy's importance in environmental behavior.

All analyses were conducted using SPSS version 28.0. Descriptive statistics, correlation analyses, and hierarchical multiple regression were performed following established protocols. Moderation analyses employed the PROCESS macro v4.0 [103] with bootstrap confidence intervals (n = 5000). Missing data were handled using listwise deletion (<2% missing cases).

## Analysis and findings

### Descriptive analysis

To evaluate how the respondents reacted to the plastic tax, we conducted a frequency analysis of the statement, "I support the idea of taxing plastic users more to reduce plastic use." Fig 1 illustrates the results. With 51.3% in favor of the idea and 13% against it, the plastic tax is found to have a strong positive sentiment among respondents. However, **35.8%** of the respondents remained neutral, indicating the potential for attitudinal change.

**Table 1. Measurement questions and reliability analysis results.**

| Measurement Concepts | | Statements | Cronbach α | scale |
|---|---|---|---|---|
| Willingness to pay plastic tax | | I favor the idea of taxing plastic users more to reduce plastic use. | .635 | 5 |
| | | I am in favor of taxing plastic users to reduce plastic use, provided that the taxes are reinvested in environmental improvements. | | |
| Value Factors | Environmentalism | I think a lot about the damage we do to the environment. | .761 | 5 |
| | | I often worry about the effects of pollution on me and my family. | | |
| | Altruism | Peaceful world: life without war and conflict | .837 | 7 |
| | | Social justice: correcting injustice, helping the weak | | |
| | | Helping: working for the well-being of others | | |
| | Egalitarianism | We need systemic reforms to share wealth equitably.. | .761 | 5 |
| | | Many of our problems would be solved if our society were more equal. | | |
| | Economic materialism | Economic development should be prioritized over conservation. | .820 | 5 |
| | | Economic development should come first, followed by environmental protection. | | |
| Risk Perception Paradigm | Perceived risk | Climate change is a very serious problem that cannot be compared to any other risk. | .798 | 5 |
| | | The changes caused by climate change will cause a lot of damage to me and my family. | | |
| | Knowledge | I am familiar with government policies to address climate change. | .744 | 5 |
| | | I am more knowledgeable than others about climate change issues. | | |
| | Government trust | The government's goals to address climate change will be successful.. | .794 | 5 |
| | | The government's goals to address climate change are achievable. | | |
| | | I believe the government is doing a good job in addressing climate change. | | |
| | Emotions | I have negative thoughts about climate change. | .726 | 5 |
| | | The picture of our future under climate change is very dark. | | |
| Environmental Psychology | Climate change skepticism | The problem of climate change has been exaggerated. | .891 | 5 |
| | | The damage caused by climate change is exaggerated. | | |
| | Guilt | I regret not doing more to prevent climate change. | .756 | 5 |
| | | I feel guilty for not taking action to address climate change.. | | |
| | Efficacy | Climate change is solvable depending on what I do. | .840 | 5 |
| | | I can solve the climate change problem if I try. | | |

The second statement about the plastic tax ("I am in favor of taxing plastic users as long as the taxes are reinvested in environmental improvements to reduce plastic use") was analyzed under the condition that the revenue would be used for environmental purposes.

Overall, **54.9% of the respondents were in favor**, while **12.1% were opposed**. Compared with Fig 1, the percentage of favorable responses increased by **3.6%**, neutral responses decreased by **2.7%**, and opposing responses slightly decreased by **0.9%**.

Although these differences are relatively small, the consistently high proportion of neutral responses(**33.1%**) suggests a lack of strong consensus regarding the plastic tax, even when its environmental benefits are emphasized (Fig 2).

To analyze the differences in plastic tax preferences according to demographic variables, we conducted a mean analysis. Fig 3 shows the results of the analysis. The average of the scores for the two questions measuring plastic tax was estimated. First, the sex-based difference showed that women had more positive attitude toward the plastic tax than men, reflecting women's greater sensitivity to environmental issues [104,105]. However, this difference between groups is not

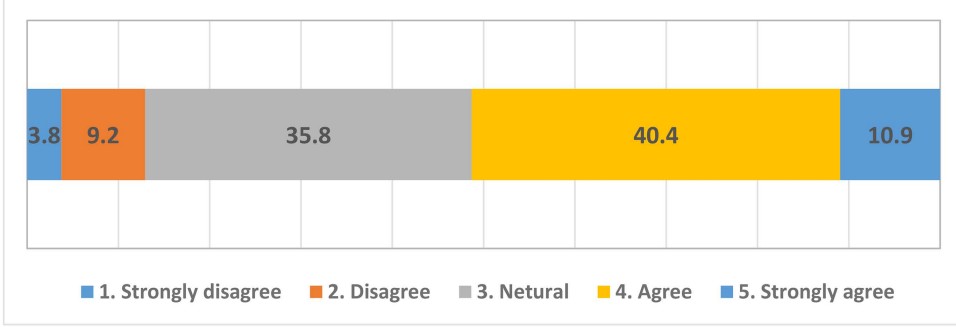

**Fig 1. Frequency of support for the plastic tax.**

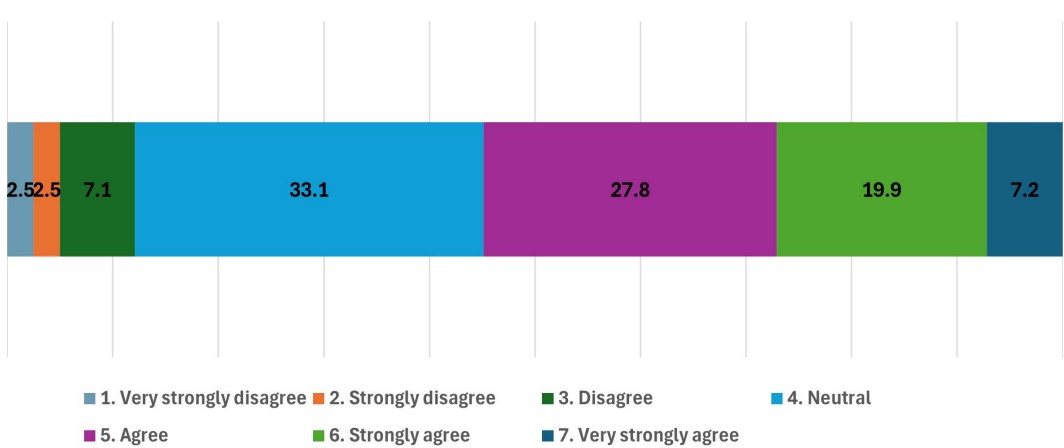

**Fig 2. Frequency of support for the plastic tax.**

significant (t = 2.134, p = 1.44). In terms of age, a positive attitude toward the plastic tax was higher among those aged 40 and older than those in their 20s and 30s, and the level of favorability increases with increasing age from 40 to 60 (f = 19.616, p = .000). This may reflect the fact that older people are more likely to have experienced the damaging effects of plastic. This may also be due to their financial ability to pay the plastic tax. Regarding education level, respondents with a high school diploma or higher were more likely to be in favor of the plastic tax than those with a high school diploma or less (t = 2.797, p = 0.095). This is the result of increased knowledge about plastic through education. Regarding income, the support for a tax is higher in the higher income group (f = 4.371, p = 0.013), probably due to the shared monetary nature of tax and income.

A bivariate correlation analysis was conducted between risk perception, value, and environmental psychological factors regarding the acceptance of the plastic tax (Table 2). Pearson's correlation coefficient has a value range from −1–1. As there is no absolute criterion to indicate whether the correlation between variables is large or small, this study considered the absolute value of 0.5 or more to represent a high correlation between variables. The results showed that perceived risk and emotion had correlation coefficients of 0.5 or higher (r = .571, p < .001).

When analyzing the support for taxes and value factors, we found that environmentalism, altruism, and egalitarianism are positively related to support for taxes, and materialism was negatively related to support for taxes. In particular, support for taxes had the highest correlation with environmentalism, followed by altruism and egalitarianism. Environmental

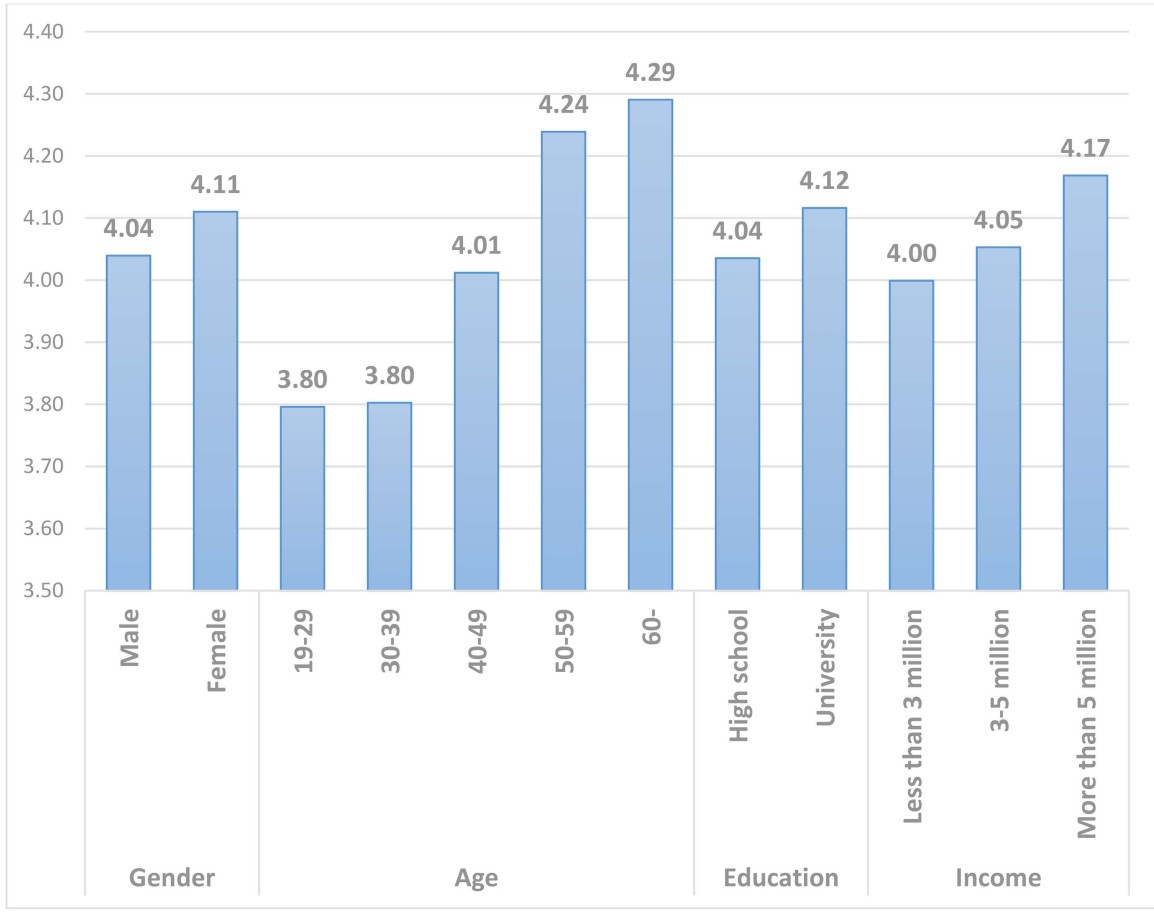

**Fig 3. Mean analysis according to demographic variables.**

**Table 2. Correlation analysis between risk perception, value, and environmental psychological factors regarding the acceptance of the plastic tax.**

| | | 1 | 2 | 3 | 4 | 5 | 6 | 7 | 8 | 9 | 10 | 11 |
|---|---|---|---|---|---|---|---|---|---|---|---|---|
| 1. Placstic Policy Support | | 1 | | | | | | | | | | |
| Value factor | 2. Environmentalism | .410*** | 1 | | | | | | | | | |
| | 3. Altruism | .334*** | .358*** | 1 | | | | | | | | |
| | 4. Egalitarianism | .187*** | .128*** | .198*** | 1 | | | | | | | |
| | 5. Materialism | −.213*** | −.194*** | −.249*** | −.023 | 1 | | | | | | |
| Risk perception factor | 6. Perceived risk | .481*** | .464*** | .404*** | .128*** | −.311*** | 1 | | | | | |
| | 7. Knowledge | .283*** | .328*** | .053* | .088*** | .099*** | .211*** | 1 | | | | |
| | 8. Trust | .364*** | .305*** | .184*** | .138*** | .048 | .306*** | .485*** | 1 | | | |
| | 9. Emotions | .331*** | .312*** | .274*** | .114*** | −.213*** | .571*** | .166*** | .168*** | 1 | | |
| Planned behavior factor | 10. Climate change skepticism | −.200*** | −.165*** | −.258*** | .013 | .385*** | −.351*** | .261*** | .142*** | −.250*** | 1 | |
| | 11. Guilt | .352*** | .386*** | .163*** | .138*** | −.051* | .353*** | .471*** | .436*** | .275*** | .096*** | 1 |
| | 12. Efficacy | .389*** | .398*** | .238*** | .150*** | −.087** | .409*** | .452*** | .545*** | .254*** | .025 | .562*** |

values had a strong correlation with the acceptance of the plastic tax because these values refer to environmental protection, nature preservation, and restoration to combat the environmental crisis. Understanding how much the planet is suffering is based on the altruistic value of caring about other living things. In this sense, altruistic values have a defining relationship with the support for the plastic tax.

Conversely, materialism had a negative relationship with the acceptance of taxes, probably due to the focus on economic development and the satisfaction of needs, in contrast to post-materialist values based on self-actualization, which links to environmentalism, altruism, and egalitarianism. These findings suggest conflicts between altruistic, egalitarianism, post-materialistic values and egoism, individualism, and materialism regarding the plastic tax.

When analyzing the support factors and risk perception paradigm, we found that perceived risk, knowledge, trust, and emotion were all positively related to support for taxes. In particular, support for taxes and perceived risk had the largest coefficient, followed by trust > emotion > knowledge. These results suggest that the risk of using plastic is a strong argument in favor of the tax.

When analyzing the support for taxes and environmental psychological factors, climate change skepticism was found to have a negative relationship with the acceptance of taxes, and guilt and efficacy had a positive relationship with the support for taxes. In particular, the acceptance of taxes and efficacy (r = .390, p < .001) had the largest coefficient value, followed by guilt and climate change skepticism. These results suggest that individuals' belief that they can do something about climate change is linked to tax. In reality, the presence of resources or individual abilities increases their sense of control over an object, thus driving relevant behavior.

The variables that are associated with the support for taxes, the most were perceived risk > environmentalism > efficacy > trust, suggesting that an individual's perceived risk of climate change is an important factor affecting policy-related tax (dis)agreement.

## Regression analysis

Multiple regression analysis was conducted to identify the factors associated with the acceptance of the plastic tax. Model 1 examined the relationship between independent variables and the support for the plastic tax without including the planned behavior factor, and Model 2 included variables in planned behavior, such as climate change skepticism, guilt, and efficacy (Table 3). The explanatory power of Model 1 was 34.2%, with a model fit of 64.373 (p < .001). The tolerance was greater than 0.1, and the VIF was less than 10, indicating no multicollinearity.

The results of Model 1 showed that, in terms of sociodemographic factors, age and education variables were significantly associated with the support for the plastic tax. A higher age and education were likely to be associated with a higher the support for the plastic tax because individuals in this group were more likely to experience the problems of climate change. They indicate the need to gain the sympathy of the young and less-educated groups because they have negative attitude toward raising taxes related to climate change in the future.

Next, among the value factors, environmentalism, altruism, and egalitarianism were significantly positively associated with the acceptance of the plastic tax, while materialism was significantly negatively associated with the support for the plastic tax. In the risk perception paradigm, perceived risk, knowledge, trust, and emotion all contributed significantly positively to the acceptance of the plastic tax. Based on the standardized regression coefficients, the variables that are associated with the support for the plastic tax were perceived risk > trust > environmentalism > knowledge > altruism > materialism > egalitarianism > emotion. Perceived risk had the strongest association with the support for the plastic tax, which means that the higher an individual's perceived risk of climate change, the higher their support for the plastic tax. In other words, the higher the perceived risk, the more willing individuals are to pay their taxes to combat climate change. We also observed that in terms of perception, trust and knowledge were operationalized, while in terms of value, environmentalism and altruism had a significant association with the support for the plastic tax. These results indicate that plastic tax

**Table 3. Regression analysis to identify factors associated with the acceptance of the plastic taxes.**

| Variables | | Model 1 | | | Model 2 | | |
|---|---|---|---|---|---|---|---|
| | | b | S.E | Beta | b | S.E | Beta |
| (Constants) | | .135 | .239 | | .478 | .249 | |
| Sociodemographic factors | Sex | −.009 | .041 | −.005 | −.038 | .040 | −.020 |
| | Age | .054*** | .015 | .082 | .054*** | .015 | .083 |
| | Education | .105* | .042 | .056 | .098* | .042 | .052 |
| | Income | .029 | .024 | .026 | .029 | .024 | .026 |
| Independent 1 Values factors | Environmentalism | .174*** | .034 | .131 | .136*** | .034 | .103 |
| | Altruism | .094*** | .023 | .099 | .082*** | .023 | .087 |
| | Egalitarianism | .101*** | .023 | .095 | .098*** | .023 | .093 |
| | Materialism | −.105*** | .025 | −.097 | −.072** | .025 | −.067 |
| Independent 2 Risk Perception Paradigm | Perceived risk | .269*** | .036 | .217 | .200*** | .037 | .161 |
| | Knowledge | .126*** | .031 | .103 | .119*** | .033 | .098 |
| | Trust | .184*** | .033 | .139 | .149*** | .036 | .113 |
| | Emotion | .071* | .032 | .058 | .053 | .032 | .043 |
| Moderating Environmental Psychological factors | Climate change skepticism | | | | −.123*** | .026 | −.122 |
| | Guilt | | | | .101** | .034 | .081 |
| | Efficacy | | | | .086* | .034 | .072 |
| F value | | 64.373*** | | | 55.503*** | | |
| R | | 0.585 | | | 0.600 | | |
| $R^2$ | | 0.342 | | | 0.360 | | |
| adj. $R^2$ | | 0.337 | | | 0.353 | | |

* p < .05, ** p < .01, *** p < .001.

acceptance requires multi-faceted policy approaches addressing both risk communication and value-based messaging. Specifically, policymakers should combine evidence-based risk communication strategies to enhance perceived climate threats with targeted messaging that resonates with environmental and altruistic values while addressing materialistic concerns through economic framing.

The explanatory power of Model 2, which adds the three variables in planned behavior to Model 1, was 36.0%, with a model fit of 55.503 (p < .001). The tolerance was above 0.1, and the VIF was below 10, indicating no multicollinearity. In Model 2, the sociodemographic factors were the same as in Model 1: Higher age and education were significantly associated with the support for the plastic tax. The value factors of environmentalism, altruism, and egalitarianism had a significantly positive effect on the support for the plastic tax, while materialism demonstrated a significantly negative relationship with the support for the plastic tax. As for the risk perception paradigm factors, perceived risk, knowledge, and trust, excluding emotion, all had a significantly positive association with the support for the plastic tax. Among planned behavior factors, climate change skepticism had a significantly negative link with the support for the plastic tax; guilt and efficacy had a significantly positive association with the support for the plastic tax.

Based on the standardized regression coefficients in Model 2, we found that the effect size is positively associated with in the order of perceived risk > climate change skepticism > trust > environmentalism > knowledge > egalitarianism > altruism > guilt > efficacy > materialism.

As in Model 1, perceived risk had the strongest positive association with the support for the plastic tax. In Model 2, the moderator variable of climate change skepticism had the second largest relationship, which can be interpreted as a result of different political frameworks. Despite the emphasis on reducing greenhouse gas emissions, doubts about

global warming have increased, for example, due to the U.S. withdrawal from the Paris Agreement in 2017. Many Americans believe that the problem of climate change has been exaggerated, meaning that climate change skepticism is widespread. As the study of climate change is a risk analysis based on scientific uncertainty, global warming has long been a matter of debate.

Thus, the results showed that, except for emotions, value, risk perception, and planned behavior variables are all significantly associated with the support for the plastic tax. In the case of the plastic tax, risk perception and skepticism about climate change play a particularly important role.

Hypothesis testing results systematically support our theoretical framework (Table 3). Value-based hypotheses were largely confirmed, validating Stern's [14] value-belief-norm theory applications to environmental policy contexts. Environmentalism significantly predicted plastic tax acceptance (H1: β = .103, p < .001), consistent with previous research demonstrating environmental values' positive impact on policy support [27,55]. This finding aligns with Mozumder's [57] work showing environmental consciousness effects on renewable energy support and extends Kim's [106] findings on environmentalist values supporting environmental protection policies to plastic taxation contexts.

Altruism showed significant positive effects (H2: β = .087, p < .001), supporting VBN theory predictions that altruistic values drive pro-environmental behaviors [14]. This result resonates with Lee et al.'s [107] demonstration that altruistic value orientation influenced eco-friendly food purchasing intentions, with altruistic values showing stronger effects than individualistic orientations. Our findings extend this pattern to environmental taxation acceptance.

Egalitarianism positively predicted acceptance (H3: β = .093, p < .001), confirming cultural theory predictions [62] that egalitarian individuals demonstrate greater sensitivity to environmental risks and support collective action. This aligns with Kahan et al.'s [24] findings that individuals with egalitarian values tend to accept climate change risks, while those with individualistic values remain skeptical.

Materialism showed expected negative effects (H4: β = −.067, p < .01), supporting post-materialist theory [64] and extending Gelissen's [67] findings on dematerialization's positive effects on environmentalism. This result confirms Hurst et al.'s [68] meta-analysis demonstrating materialism's negative correlation with environmental attitudes and validates Kilbourne et al's [70] argument that materialistic values predispose individuals toward less supportive environmental policy attitudes.

Risk perception hypotheses received strong empirical support, validating psychometric paradigm applications [108]. Perceived risk emerged as the strongest predictor (H5: β = .161, p < .001), consistent with Kim et al's [74] findings that higher perceived climate change risk increases response behavior intensity. This supports Visschers et al's [72] demonstration that perceived risks significantly influence acceptance decisions across various technologies.

Knowledge significantly predicted acceptance (H6: β = .098, p < .001), supporting O'Connor et al.'s [49] findings that greater knowledge associates with higher climate action intentions. However, our results acknowledge Bamberg et al's [23] caveat that knowledge functions as necessary but insufficient condition within broader psychological processes, requiring combination with values and situational factors [77].

Trust showed substantial effects (H7: β = .113, p < .001), validating trust's role as representative socio-cultural factor in risk research [82]. This finding extends Visschers et al's [72] work on government trust in nuclear power operations to climate policy contexts, supporting our competence trust measurement approach [36].

Planned behavior factors demonstrated expected patterns, supporting TPB applications to environmental contexts [16]. Climate skepticism showed strong negative effects (H9: β = −.122, p < .001), consistent with environmental skepticism research showing overestimated crisis beliefs and data error concerns [74]. Guilt demonstrated positive effects (H10: β = .081, p < .01), supporting Ferguson et al's [90] findings that collective guilt mediates climate change beliefs and mitigation behaviors. Efficacy showed positive associations (H11: β = .072, p < .05), extending Bandura's [93] self-efficacy framework to environmental policy contexts and confirming Lee et al.'s [94] findings on efficacy's positive effects on eco-friendly behaviors. The associations in this study should be interpreted as correlational relationships rather than causal effects, given our cross-sectional research design."

## Interaction effect analysis

Following the previous analysis, this study analyzed the moderating effects of the environmental psychological factors of climate change skepticism, guilt, and efficacy. Baron and Kenny's [109] three-step test was used to check the variables and interaction terms, and the results were tested for the simple slope effect. The results showed that climate change skepticism, a moderating variable, had an interaction effect with materialism and emotion, which are independent variables. Table 1 in S1 Appendix in support information lists the results of the interaction effect analysis.

Figs 4 and 5 show climate change skepticism as a moderating variable. Fig 4 shows that as materialism increases, support for the plastic tax decreases. However, this effect is reinforced in groups with low climate change skepticism. In the high climate change skepticism group, increased materialism does not increase the support for the tax. These results suggest that climate change skepticism and materialism interact to create a negative attitude toward plastic taxes. Climate skepticism significantly moderated materialism-acceptance relationships, where materialistic individuals under high skepticism conditions showed particularly strong negative attitudes toward plastic taxes. This interaction supports motivated reasoning theory [24], suggesting that skeptical individuals use materialistic values to justify opposition to environmental policies.

Fig 5 shows that negative emotion, a component of the risk perception paradigm, interacts with climate change skepticism.. The skepticism-emotion interaction indicates that negative climate emotions have stronger positive effects on tax support among highly skeptical individuals, potentially representing cognitive dissonance resolution mechanisms. This finding extends Slovic et al.'s [83] work on emotional responses to environmental risks and supports Kim et al's [82] findings on negative emotions' significant impacts on nuclear energy acceptance.

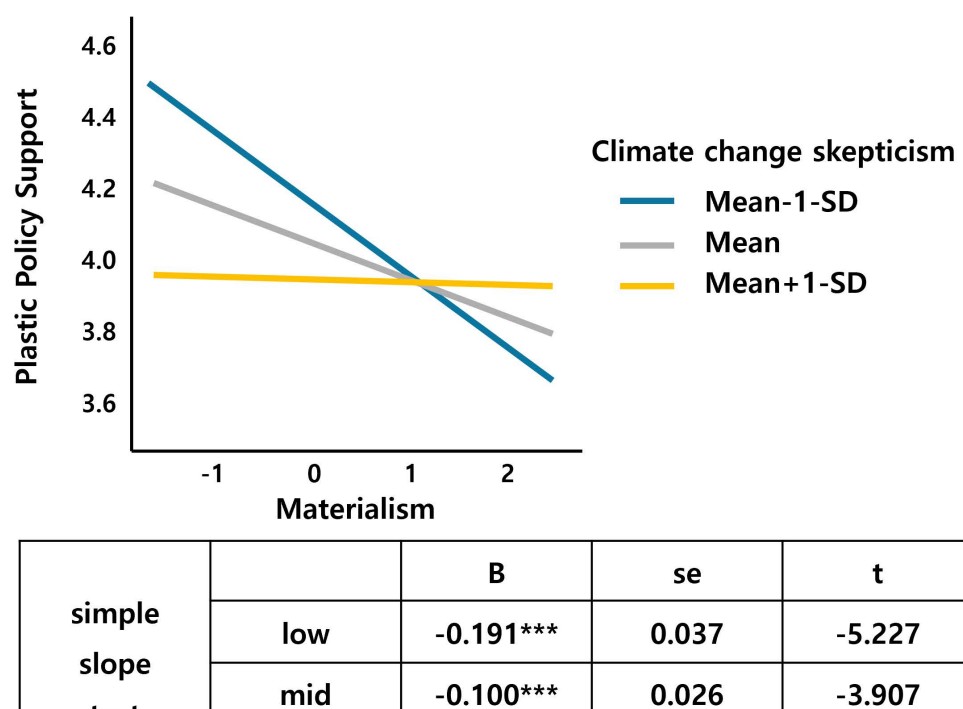

|  |  | B | se | t |
|---|---|---|---|---|
| simple slope test | low | -0.191*** | 0.037 | -5.227 |
|  | mid | -0.100*** | 0.026 | -3.907 |
|  | high | -0.010 | 0.031 | -0.312 |

**Fig 4. Materialism*Climate change skepticism.**

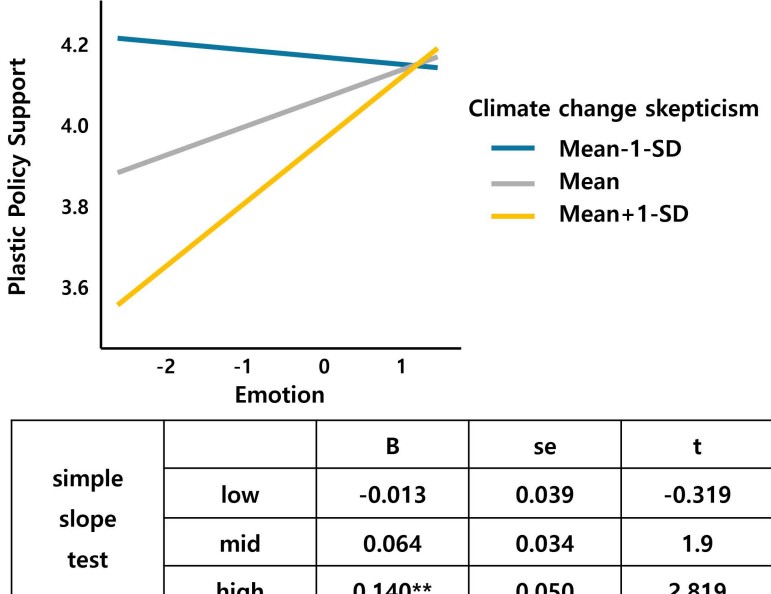

| | | B | se | t |
|---|---|---|---|---|
| simple slope test | low | -0.013 | 0.039 | -0.319 |
| | mid | 0.064 | 0.034 | 1.9 |
| | high | 0.140** | 0.050 | 2.819 |

**Fig 5. Emotion*Climate change skepticism.**

Next, guilt as the moderator variable interacts with environmentalism, altruism, egalitarianism, and perceived risk (Table 2 in S1 Appendix in support information).

Fig 6 shows that environmentalism, a value factor, interacts with guilt. As environmentalism increases, the support for the plastic tax increases, which is larger for those who feel more guilty about climate change than for those who feel less guilty. Altruism also interacts with guilt, and an increase in altruism leads to an increase in the support for the plastic tax, with a larger slope for the middle- and high-guilt groups than for the low-guilt groups (Fig 7). Fig 8 shows that the support for the plastic tax increases as egalitarianism increases, with the width of the slope for guilt increasing in the low and middle groups compared to the high group. Fig 9 shows an interaction effect between perceived risk and guilt in the risk perception paradigm. As the perceived risk increases, the support for the plastic tax increases, with a larger slope for those with higher guilt group.

Guilt demonstrated multiple significant interactions, moderating environmentalism ($\beta = .053$, $p < .05$), perceived risk ($\beta = -.088$, $p < .05$), altruism ($\beta = .050$, $p < .05$), and egalitarianism ($\beta = -.050$, $p < .05$) effects. These patterns suggest emotional amplification mechanisms where guilt enhances value-based motivations while potentially overwhelming risk-based reasoning, consistent with affective-cognitive models of environmental behavior [23]. Guilt's multiple interactions demonstrate emotional amplification mechanisms predicted by affect-cognition theories [29].

Table 3 in S1 Appendix in support information demonstrates that the moderating variable efficacy has an interaction effect with altruism and perceived risk. Fig 10 shows that as altruism increases, the support for the tax increases, and the slope is larger for the middle- and high-efficacy groups than for the low-efficacy groups. Finally, Fig 11 shows that the support for the tax increases as perceived risk increases, with a larger slope for the high-efficacy group than for the low-efficacy group. The regression analysis shows that the perceived risk of climate change has a significant effect on the acceptance of plastic textiles, which can be moderated by self-efficacy. Efficacy interactions with altruism ($\beta = .065$, $p < .01$) and perceived risk ($\beta = -.083$, $p < .05$) indicate that self-efficacy beliefs enhance altruistic motivations while moderating risk perception effects. This supports Bandura's [93] framework showing efficacy's influence on cognitive functioning through self-satisfaction with personal development and goal achievement.

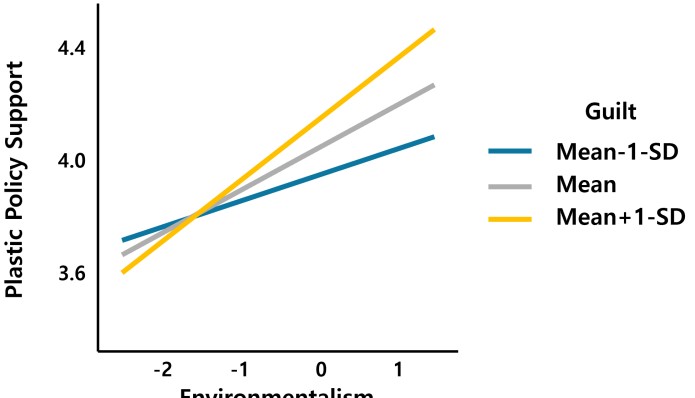

| simple slope test | | B | se | t |
|---|---|---|---|---|
| | low | 0.117** | 0.041 | 2.870 |
| | mid | 0.174*** | 0.035 | 5.000 |
| | high | 0.231*** | 0.049 | 4.700 |

**Fig 6. Environmentalism*Guilt.**

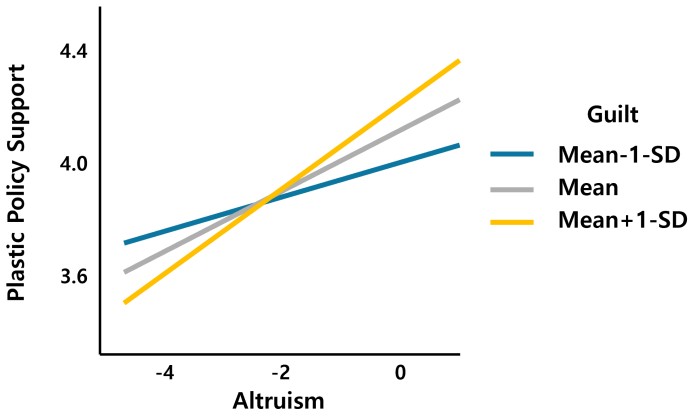

| simple slope test | | B | se | t |
|---|---|---|---|---|
| | low | 0.057 | 0.030 | 1.900 |
| | mid | 0.100*** | 0.023 | 4.350 |
| | high | 0.143*** | 0.033 | 4.320 |

**Fig 7. Altruism*Guilt.**

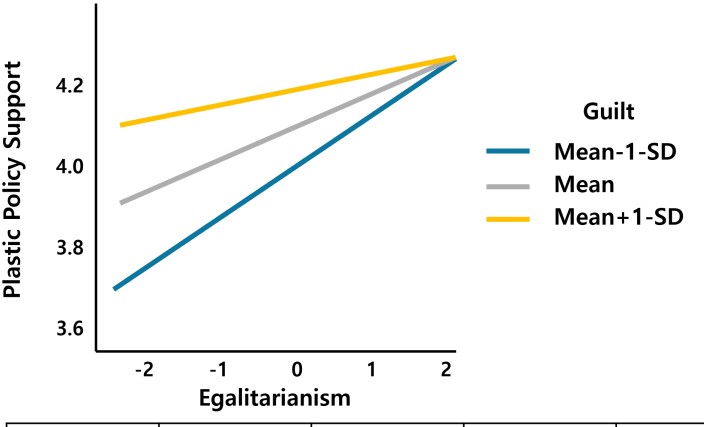

| simple slope test | | B | se | t |
|---|---|---|---|---|
| | low | 0.133*** | 0.031 | 4.300 |
| | mid | 0.085*** | 0.023 | 3.670 |
| | high | 0.037 | 0.032 | 1.170 |

**Fig 8. Egalitarianism*Guilt.**

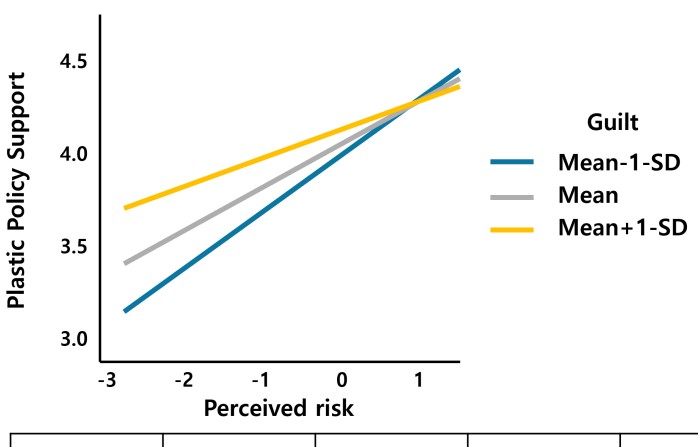

| simple slope test | | B | se | t |
|---|---|---|---|---|
| | low | 0.319*** | 0.046 | 6.91 |
| | mid | 0.238*** | 0.037 | 6.51 |
| | high | 0.156** | 0.052 | 3.02 |

**Fig 9. Perceived risk*Guilt.**

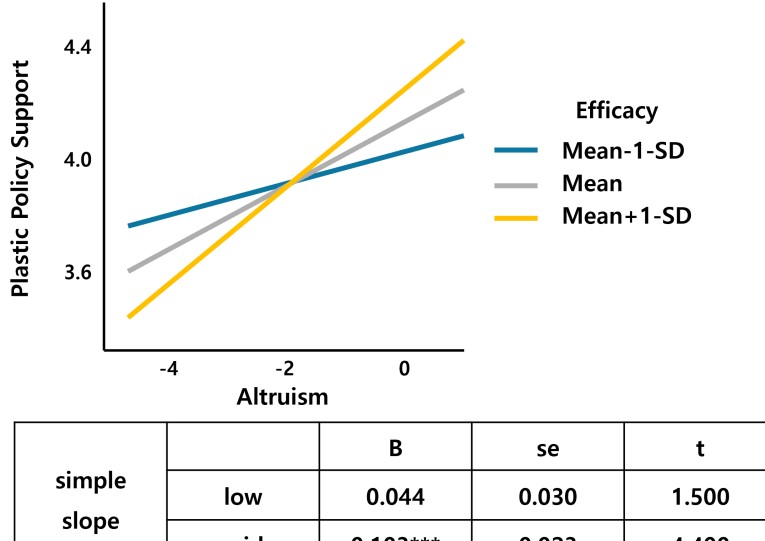

| simple slope test | | B | se | t |
|---|---|---|---|---|
| | low | 0.044 | 0.030 | 1.500 |
| | mid | 0.103*** | 0.023 | 4.400 |
| | high | 0.162*** | 0.035 | 4.600 |

**Fig 10. Altruism*Efficacy.**

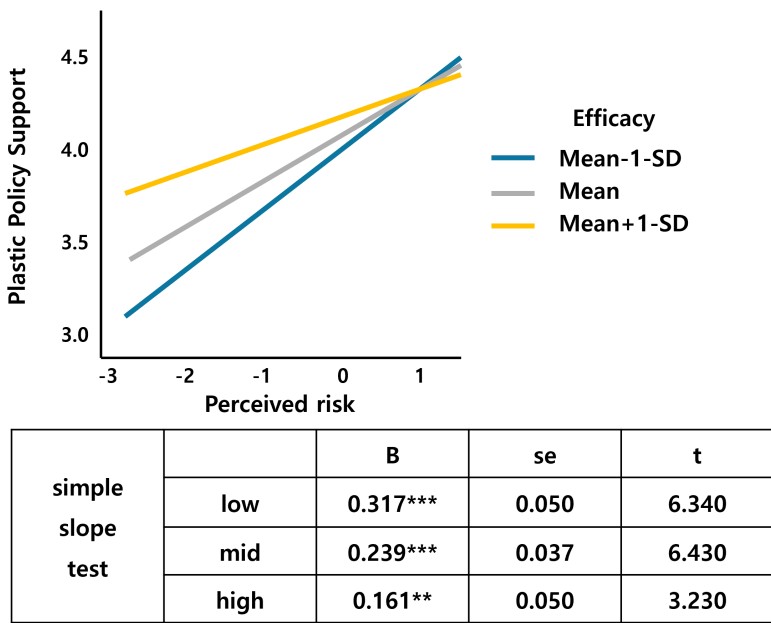

| simple slope test | | B | se | t |
|---|---|---|---|---|
| | low | 0.317*** | 0.050 | 6.340 |
| | mid | 0.239*** | 0.037 | 6.430 |
| | high | 0.161** | 0.050 | 3.230 |

**Fig 11. Perceived risk*Efficacy.**

## Results and summary

The negative effects of climate change are a current cause for concern, rather than a future issue. In addition, the convenience of consumption exacerbates the problem of climate change. In particular, the unconscious use of plastics has emerged as a leading cause of the increase in greenhouse gases, emphasizing the urgent need to regulate plastics. Thus, introducing a plastic tax is essential; however, the acceptability of the tax seems to hinder its implementation. This study analyzed the determinants of the decision to pay a tax to reduce plastic use by setting the planned behavior factors such as climate change skepticism, guilt, and efficacy as predictor and moderating variables. The main findings of this study are summarized as follows.

First, in terms of value factors, environmentalism, altruism, and egalitarianism had a significantly positive effect on the support for the plastic tax, whereas materialism had a significantly negative effect. In addition, in the risk perception paradigm, perceived risk, knowledge, and trust had significantly positive effects on the support for the tax. Regarding the planned behavior factors, climate change skepticism, guilt, and efficacy had significant effects on the support for the plastic tax, with climate change skepticism demonstrating a negative effect and guilt and efficacy showing a positive effect.

The explanatory power of the variables in the factors mentioned above increased in the order of perceived risk > climate change skepticism > government trust > environmentalism > knowledge > egalitarianism > altruism > guilt > efficacy > materialism, suggesting that perceived risk plays a crucial role in the support for the plastic tax.

This study examined the interaction effects of value and risk perception paradigm factors with the environmental psychological factors of climate change skepticism, guilt, and efficacy as moderating variables. The results showed that climate change skepticism interacted with materialism and emotion, and guilt interacted with environmentalism, altruism, egalitarianism, and perceived risk. Finally, we found that efficacy interacted with altruism and perceived risk. In particular, when analyzing the effects of personal values and risk perception paradigm factors on the support for the tax, we found that planned behavior acted as a moderating variable.

Based on the above results, among the 11 hypotheses, environmentalism (Hypothesis 1), altruism (Hypothesis 2), egalitarianism (Hypothesis 3), materialism (Hypothesis 4), perceived risk (Hypothesis 5), knowledge (Hypothesis 6), trust (Hypothesis 7), and emotion (Hypothesis 8) were accepted, and emotion hypothesis was rejected. In addition, the moderating effect of climate change skepticism (Hypothesis 9) on materialism and emotion was partially accepted, that of guilt (Hypothesis 10) on environmentalism, altruism, egalitarianism, and perceived risk was partially accepted, and that of efficacy (Hypothesis 11) on altruism and perceived risk was partially accepted.

## Discussion and implications

We analyzed factors determining plastic tax support, focusing on willingness to pay rather than tax amount, finding that values, risk perception, and planned behavior factors determine support for plastic taxes.

Our study demonstrates that effective plastic tax policies must simultaneously address values, risk perceptions, and planned behavior elements. This integrated approach aligns with van der Werff et al. [110], who showed environmental policies succeed when addressing both motivational and implementation factors. However, balancing these factors faces challenges from potential conflicts. Gifford [111] notes that policies targeting rational risk assessments may clash with deeply held values, while Steg et al. [112] emphasize that behavioral control interventions can be ineffective if underlying values are unaddressed. Additionally, Harring et al. [113] demonstrate that policies focusing on economic incentives may undermine intrinsic environmental motivations if perceived as coercive.

### Theoretical insights

Our findings provide theoretical insights into plastic tax acceptance mechanisms. The strong predictive power of perceived risk ($\beta = .161$) supports psychometric paradigm assertions that subjective risk evaluations drive policy preferences more than objective assessments [108], extending Kahan et al.'s [24] work on motivated reasoning where individuals process climate information through value-laden filters.

Value-based patterns align with Stern's [14] VBN theory, where environmentalism, altruism, and egalitarianism activate moral obligation toward collective action. Our study uniquely demonstrates that materialistic values create competing motivational frameworks, supporting post-materialist theory [64] while extending it to taxation contexts.

Climate skepticism's strong negative effect ($\beta = -.122$) and moderation patterns reveal motivated reasoning processes where ideological beliefs filter information processing [24]. Our contribution lies in demonstrating how skepticism interacts with materialism to strengthen opposition, suggesting policy communication must address both cognitive and motivational barriers simultaneously.

The guilt and efficacy interactions represent theoretical advances in environmental psychology. Unlike previous studies focusing on direct effects [90], our research reveals boundary conditions where emotional and cognitive factors amplify or attenuate value-based motivations.

## Policy implications

These findings have important policy implications requiring simultaneous consideration of multiple variables for plastic tax implementation. From a factor perspective, balanced mixes of values, risk perception, and planned behavior should be reflected in policy.

First, introducing plastic tax requires educating the public about climate change risks. Although climate change risks are visible, risk communication remains lacking. Therefore, we need to effectively communicate facts and present climate change evidence to the public.

Second, environmentalism, trust, and climate change concerns have significant relationships with plastic tax acceptability. These variables differ in nature: environmentalism relates to values, trust to perceptions, and climate change skepticism to evaluative attitudes. Persuasive strategies should consider these attributes through three approaches: values-based persuasion using moral reframing strategies that align messages with recipients' core values [114]; perception-focused strategies using visualization and narrative techniques making abstract risks concrete [115]; and attitude-targeted interventions leveraging existing positive attitudes as bridges to new behaviors [116].

Our main finding that climate change skepticism, guilt, and efficacy act as moderating variables suggests plastic tax implementation requires strategic efforts focused on overcoming skepticism, imposing responsibility, and increasing efficacy. Skepticism can be addressed through evidence-based communication strategies as demonstrated in Ireland's successful plastic bag levy [117]. For imposing responsibility, the UK's approach combined guilt messaging with corporate responsibility frames, resulting in 86% reduction in single-use plastic bags [118]. Efficacy was effectively built in Sweden through community-based initiatives showcasing tangible impacts [119].

Our findings confirmed the policy-relevant mechanism connecting public acceptance to plastic tax implementation and consumption reduction. Nielsen et al. [22] demonstrates that public acceptance is critical for successful plastic tax implementation, minimizing political resistance and increasing compliance. When implemented, plastic taxes effectively reduce consumption – Homonoff et al. [11] documented 42% decrease in plastic bag use after taxation in Chicago, while Convery et al. [117] that Ireland's plastic bag levy led to a reduction in usage by over 90%, demonstrating how strong public support enabled effective implementation and substantial behavioral change.

## Implementation recommendations

To strengthen policy relevance, we offer specific recommendations. First, targeted communication strategies addressing psychological factors are essential. Rivers et al. [10] demonstrated that messages emphasizing collective efficacy increased environmental tax support by 37%, while Gifford et al. [111] showed guilt-framed messages were effective for environmentally-concerned citizens but counterproductive for skeptics. Phased implementation with initial focus on single-use items allows adaptive policy design. Xanthos et al. [120] found gradual introduction with 6–12 month preparation periods substantially improved business compliance and public acceptance.

Second, policymakers can strategically frame plastic tax initiatives to align with dominant cultural values rather than attempting to change them. Bolderdijk et al. [121] demonstrated that environmental policies framed to resonate with existing public values achieve higher acceptance rates. Communication strategies can emphasize how plastic taxes support specific value orientations—highlighting environmental protection for those with strong environmentalism values or economic efficiency for those with materialist values.

## Future research and limitations

Future research should extend this framework across multiple domains. Testing our integrated model with other environmental policies would establish generalizability beyond plastic taxation. Cross-cultural validation is essential, as cultural values significantly influence environmental policy acceptance [17]. Experimental designs could establish causal relationships through field experiments manipulating risk communication messages or laboratory studies varying guilt induction. Longitudinal designs tracking attitude changes over policy implementation periods would reveal dynamic relationships between values, perceptions, and support [122]. Investigating policy sequencing effects would inform strategic policy implementation, addressing spillover effects in environmental behavior change [123].

This study has several important limitations. Our cross-sectional design precludes causal inferences about relationships between psychological factors and policy support. Self-reported measures may suffer from social desirability bias, particularly regarding environmental attitudes [124]. The single-country Korean context limits generalizability across cultural and political systems. Brief scales (2–3 items) may inadequately capture complex constructs like guilt and efficacy, and we measured behavioral intentions rather than actual tax-paying behavior, creating potential intention-behavior gaps [122].

A significant limitation is our failure to measure participants' political ideology and party affiliation. Since plastic taxation represents environmental policy with political implications, political orientations likely substantially influence support attitudes. Research demonstrates that political ideology strongly predicts environmental policy acceptance, with conservatives showing greater skepticism toward climate policies than liberals [102]. Political partisanship also affects how individuals process environmental information, potentially moderating relationships between psychological factors and policy support [24].

This study's Korean cultural context significantly limits generalizability. Environmental values and policy preferences vary substantially across cultures [125]. Korean collectivistic values and hierarchical social structures may influence plastic tax acceptance differently than individualistic Western contexts [126]. Our findings specifically reflect Korean environmental attitudes shaped by rapid industrialization experiences and Confucian value systems emphasizing collective responsibility [65]. Cross-cultural validation is essential before generalizing results to other contexts, particularly given documented cultural variations in environmental concern and policy support [17].

Finally, the relationships identified represent associations rather than causal effects, given our cross-sectional correlational design. Future research should employ causal research designs to test the directional relationships proposed in our theoretical framework.

## Supporting information

**S1 Appendix.**
(PDF)

## Author contributions

**Conceptualization:** Seoyong Kim.

**Data curation:** Seoyong Kim.

**Formal analysis:** Miri Kim.

**Investigation:** Miri Kim.

**Methodology:** Sehyeok Jeon.

**Visualization:** Sehyeok Jeon.

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
