## [Decision Letter · Decision Letter 0]

12 Jun 2025

PONE-D-24-54075Can Climate Change Perceptions Drive Plastic Policy Support? Effects of Climate Change Skepticism, Guilt, and Efficacy on the Acceptance of the Plastic TaxPLOS ONE?

Dear Dr. Kim,

Thank you for submitting your manuscript to PLOS ONE. After careful consideration, we feel that it has merit but does not fully meet PLOS ONE’s publication criteria as it currently stands. Therefore, we invite you to submit a revised version of the manuscript that addresses the points raised during the review process.

I recommend that it should be revised taking into account the changes requested by the reviewers. Since the requested changes include valuable and constructive reviews, I would like to give you a chance to revise your manuscript. The revised manuscript will undergo the next round of review by same reviewers.

We look forward to receiving your revised manuscript.

Kind regards,

Baogui Xin, Ph.D.

Academic Editor

PLOS ONE

Journal Requirements:

“This work was supported by the Ministry of Education of the Republic of Korea and the National Research Foundation of Korea (NRF-2021S1A5C2A02087244). The Human Resources Development Project for HLW Management hosted by KORAD and MOTIE”

“This work was supported by the Ministry of Education of the Republic of Korea and the National Research Foundation of Korea (NRF-2021S1A5C2A02087244). The Human Resources Development Project for HLW Management hosted by KORAD and MOTIE”

“This work was supported by the Ministry of Education of the Republic of Korea and the National Research Foundation of Korea (NRF-2021S1A5C2A02087244). The Human Resources Development Project for HLW Management hosted by KORAD and MOTIE”

Reviewers' comments:

Reviewer's Responses to Questions

**Comments to the Author**

1. Is the manuscript technically sound, and do the data support the conclusions?

Reviewer #1: No

Reviewer #2: Partly

Reviewer #3: Yes

2. Has the statistical analysis been performed appropriately and rigorously?

Reviewer #1: No

Reviewer #2: Yes

Reviewer #3: Yes

3. Have the authors made all data underlying the findings in their manuscript fully available?

Reviewer #1: Yes

Reviewer #2: Yes

Reviewer #3: Yes

4. Is the manuscript presented in an intelligible fashion and written in standard English?

Reviewer #1: No

Reviewer #2: No

Reviewer #3: No

Reviewer #1: In its current form, the manuscript has a strong foundation but requires moderate to major revision to reach its full potential. I recommend the authors undertake the following key revisions:

Revise the introduction to better integrate the theories and justify the research questions. Clearly explain how climate change perceptions conceptually link to plastic tax support, and present specific hypotheses for each predictor (and moderator) grounded in literature.

Expand the literature review to include additional relevant studies (see Suggested References). This will strengthen the theoretical arguments and show the novelty of the work in context. Make sure to discuss how each added reference relates to your study (e.g., as precedent or contrast).

Provide more methodological clarity: detail the sample and procedure, ensure measures are described with sources and reliability, and report how the data were analyzed (regression/SEM steps, checks performed). This transparency will bolster confidence in the findings.

Reorganize and clarify results: link them back to hypotheses, report statistics comprehensively, and explain the interaction effect in an accessible way. Consider adding a figure for the moderation and a table for descriptive stats.

Deepen the discussion: interpret why the findings occurred in light of theory and prior research, highlight the study’s contributions, and acknowledge its limitations frankly. Also, suggest avenues for future research (e.g., testing these relationships in other policy domains or with experimental designs to establish causality).

Thoroughly copy-edit the language: perhaps seek a native English speaker or professional editor to polish the manuscript. Improving grammar, phrasing, and consistency will prevent misinterpretation and give the paper a professional finish.

Suggested References:

Zhang, C., Liu, L., & Xiao, Q. (2022). The influence of Taoism on employee low-carbon behavior in China: The mediating role of perceived value and guanxi. Psychology Research and Behavior Management, 15, 2169–2181. https://doi.org/10.2147/PRBM.S371945

Liu, L., Zhang, C. (2022). Linking environmental management accounting to green organisational behaviour: The mediating role of green human resource management. PLOS ONE, 17(12), e0279568. https://doi.org/10.1371/journal.pone.0279568

Zhang, C., Ma, X., & Liu, L. (2023). The effect of passion for outdoor activities on employee well-being using nature connectedness as the mediating variable and environmental identity as the moderating variable. Psychology Research and Behavior Management, 16, 4883–4896. https://doi.org/10.2147/PRBM.S436612

Dai, Q., Peng, S., Guo, Z., Zhang, C., Dai, Y., Hao, W., et al. (2023). Place identity as a mediator between motivation and tourist loyalty in ‘red tourism’. PLOS ONE, 18(10), e0284574. https://doi.org/10.1371/journal.pone.0284574

Zhang, C., & Liu, L. (2023). Exploring the role of employability: The relationship between health-promoting leadership, workplace relational civility and employee engagement. Management Decision, 61(9), 2582–2602. https://doi.org/10.1108/MD-05-2022-0679

Reviewer #2: In this paper, the authors investigate people’s acceptance of a plastic tax to reduce plastic use. Using an online survey with a representative sample of participants (N = 1571), the authors explore associations between acceptance of a plastic tax and various psychological variables, including values, perceptions of risk from climate change, scepticism about climate change, guilt, and self-efficacy. The study finds significant relationships between most of these variables, with varying effect sizes, and reports moderating roles of scepticism, guilt, and self-efficacy. The authors conclude the paper with some policy recommendations.

In general, I thought that the paper started relatively strongly, with a good discussion of the global statistics on plastic use and a useful literature review on the psychological determinants of attitudes towards climate change. However, I have a number of concerns about the paper, particularly with regards to the methods, results, and discussion sections. For context, I am not an expert on the theory of planned behaviour, so I have focused my review on the general writing and clarity of the paper and the claims made therein. The authors should deal with these comments in any revision.

COMMENTS

Quality of writing:

Unfortunately, there are several points in the paper where the writing lacked the necessary clarity and/or was grammatically incorrect. For example, the first sentence of the abstract reads: “For the circular economy, it needs for solution about plastic problems”. On line 277, the authors state that “climate change can save the plastic world”. It’s not clear what phrases like this mean. In other places, there were mistakes in the writing, e.g., “In particular, support for taxes had the highest correlation coefficient environmentalism, followed by environmentalism and altruism” (line 695). In a revision, the authors should go through the text carefully and tidy up any sentences that are grammatically incorrect or contain errors.

Causal language:

Since this is a correlational study that does not present a causal model, the results cannot speak to causality. The authors should therefore remove any causal language throughout the paper, including words like “drives” (used in the title), “impacts” (used in the abstract), “influences”, “determines”, etc. Instead, the authors should refer only to associations, relationships, and correlations. Moreover, the authors should caveat the explicit causal pathway that they suggest in the discussion section, clarifying that their data are unable to support a model like this.

Moderation predictions in introduction section:

While I thought that the different paradigms were nicely laid out in the introduction section, it was not clear to me why Hypotheses 9, 10, and 11 predicted moderating effects of the planned behaviour variables. The introduction should do a better job explaining why these variables are expected to moderate the effects of values and perceived risk. Is this predicted from previous research?

Missing details in methods section:

The methods section was missing some important details. Presumably the study sampled participants from the Republic of Korea, but this was not clearly stated. This section should also include subsections describing the procedure of the study (survey flow, randomisation of survey blocks, etc.), the statistical analyses and how they were conducted (using R, Stata, etc.), and the availability of data and analysis code. While all survey items are presented in the text, the authors might also consider moving these into a table for readability.

Concerns about ethical approval:

The paper explains why ethical approval was not obtained. I don’t have sufficient knowledge about the Korean Bioethics and Safety Act to determine whether this is ethically sound or not, so I will defer to the editor on this.

Issues with figures in results section:

There were a few issues with the figures in the main text. In Figure 1, the percentage choosing the neutral option does not match the percentage reported in the text. Which of these is correct? In Figure 2, the grey colour would be preferable for the neutral option to match with the other figure. I was also confused to see a 7-point scale in this figure, which is different from the 5-point scale described in the methods section: which is correct? In Figure 3, the y-axis should include the whole scale (1-5) and the bars should also contain standard error bars around the means.

Reporting and interpreting the results:

There were also a few issues with reporting and interpreting the results of the statistical analyses. In Table 2, I found Model 2 difficult to understand because climate change scepticism, guilt, and self-efficacy are described as moderating variables – yet they are only included as additional variables without interaction effects. My preference would be to remove this model entirely and instead move straight to the interaction effects and simple slopes in the next subsection. Also, sometimes the results were not interpreted with sufficient clarity (e.g., "these results suggest that both perceptual approaches and efforts to change the philosophy of life should be carried out simultaneously to improve the acceptability of plastic taxes"; page 37) – sentences like these should be ironed out.

Moderating effects not explained well:

The analyses find evidence for some interaction effects between “planned behaviour variables” and other psychological variables. These interactions are described in the results section, but their importance is somewhat lost on the reader. Are these interaction effects important? Do they make sense based on past research? Do they build on previous research? The results and discussion sections should do a better job explaining and interpreting these effects.

Discussion section does not clearly build on results:

In the discussion section, the authors sometimes make empirical claims or policy recommendations that are not supported by their data. For example, the authors say that “our results demonstrate knowledge as a necessary but insufficient factor in behavior change” and that “knowledge components interact with attitudinal and normative factors rather than operating in isolation” (pages 46-47). But it’s not clear if the statistical analyses actually support these claims. As another example, the authors go on to make specific policy recommendations that do not seem supported by the data (e.g., “progressive tax rate structures can address equity concerns while maximizing environmental impact”; page 47). The authors should ensure that the claims in their discussion clearly draw on the results of their study.

Issues with limitations section:

In their limitations section, the authors write that “it may have been necessary to discuss the findings in the context of TPB” (page 48), but it’s not clear why the authors didn’t actually do this. Also, I feel that this section misses a major limitation, which is that the survey did not measure participants’ political ideology and political party support. Since the outcome variable is not a behavioural measure but a measure of support for a taxation policy, political leanings are likely to be crucial in shaping people’s acceptance – see McCright & Dunlap (2011) for an example in an American context. The authors should mention this as an additional limitation.

Other minor issues:

- I wasn’t sure why the authors ticked that they had competing interests but then did not declare them – unless I’m mistaken?

- It’s not standard to refer to journal names in the text (line 594 onwards) – consider revising.

- It wasn’t clear to me how the following item was a measure of emotions: “I believe the government is doing a good job in addressing climate change” (line 629)

- For frequentist statistics, it’s standard to use lower-case and italicised t-values and p-values (e.g., line 671)

- Appendix Tables 1-3 appear to be missing asterisks for some parameters

- Section 5 is a redundant “results” section – consider moving this text to the start of the discussion section

Overall, I hope that these suggested revisions will improve the quality of this paper on an important topic. If you have any questions about these comments, feel free to get in touch with me at the email address below.

Review signed: Scott Claessens

Email: scott.claessens@gmail.com

References:

McCright, A. M., & Dunlap, R. E. (2011). Cool dudes: The denial of climate change among conservative white males in the United States. Global Environmental Change, 21(4), 1163-1172.

Reviewer #3: Manuscript: "Can Climate Change Perceptions Drive Plastic Policy Support? Effects of Climate Change Skepticism, Guilt, and Efficacy on the Acceptance of the Plastic Tax"

The manuscript has significant English language problems throughout that impede comprehension and would require substantial editing before publication. Examples include:

Abstract:

- "For the circular economy, it needs for solution about plastic problems" (grammatical error)

- "Plastic waste is choking the planet...and taking over all the land on Earth" (imprecise/dramatic language)

- Multiple issues with sentence structure

Throughout the manuscript:

- Inconsistent verb tenses

- Awkward phrasing that obscures meaning

- Non-standard academic writing conventions

- Word choice issues that affect precision

The manuscript requires comprehensive English language editing by a native speaker or professional academic editor. Current language issues significantly impede readability and comprehension, which is essential for peer review and publication.

Further, while this manuscript addresses an important and timely question about public acceptance of environmental policies, several theoretical concerns arise.

The attempt to integrate Value-Belief-Norm theory, Theory of Planned Behavior, and risk perception paradigm appears forced rather than theoretically driven. The mapping of constructs to theoretical components lacks coherence – e.g., the relationship between climate change risk perception and plastic tax support is theoretically underdeveloped. The three theoretical frameworks are treated as separate rather than truly integrated.

Provide a stronger conceptual rationale for why integration of these specific frameworks is necessary and how they complement each other.

The South Korea-only sample limits generalizability, particularly given that environmental values and policy preferences vary significantly across cultures. Discuss cultural limitations more thoroughly and avoid broad generalizations about policy implications beyond the Korean context.

There are multiple "Results" sections and this creates confusion. The manuscript would benefit from a re-structuring.

**Do you want your identity to be public for this peer review?** For information about this choice, including consent withdrawal, please see our Privacy Policy

Reviewer #1: No

Reviewer #2: **Yes: ** Scott Claessens

Reviewer #3: No

---

## [Author Response · Author response to Decision Letter 1]

20 Aug 2025

Revision Note for Reviewer 1

Reviewer #1: In its current form, the manuscript has a strong foundation but requires moderate to major revision to reach its full potential. I recommend the authors undertake the following key revisions:

□ □ Comment: Revise the introduction to better integrate the theories and justify the research questions. Clearly explain how climate change perceptions conceptually link to plastic tax support, and present specific hypotheses for each predictor grounded in literature.

○ Answer:

We acknowledge the reviewer's concern regarding the theoretical integration and conceptual linkage in the introduction. We have substantially revised the introduction to better connect the three theoretical frameworks (VBN theory, TPB, and risk perception paradigm) and provide clearer justification for how climate change perceptions relate to plastic tax support.

○ Revision:

The conceptual foundation linking climate change perceptions to plastic tax support rests on three integrated theoretical frameworks. First, the Values-Beliefs-Norms (VBN) theory demonstrates that environmental values influence policy support through awareness of consequences (Stern, 2000). Recent research shows that individuals with stronger environmental values exhibit greater willingness to support climate policies, including carbon pricing mechanisms (Fairbrother, 2022).

Second, the Theory of Planned Behavior (TPB) suggests that attitudes, subjective norms, and perceived behavioral control influence behavioral intentions (Ajzen, 1991). Climate change skepticism, as an attitudinal component, has been shown to significantly moderate environmental policy support (Hornsey et al., 2016). Guilt, representing normative pressure, motivates pro-environmental behaviors when individuals perceive personal responsibility for climate change (Bissing-Olson et al., 2016).

Third, the risk perception paradigm indicates that perceived climate risks drive support for mitigation policies (van der Linden et al., 2015). Studies demonstrate that higher climate risk perception correlates with increased support for environmental regulations (Drews & van den Bergh, 2016).

The conceptual link between climate change perceptions and plastic tax support emerges from the established connection between plastic production and greenhouse gas emissions. Research indicates that plastic lifecycle contributes approximately 4-8% of global emissions (Zheng & Suh, 2019). Citizens who perceive climate change as a serious risk are more likely to support policies targeting emission sources, including plastic taxation (Nielsen et al., 2021).

References

Ajzen, I. (1991). The theory of planned behavior. Organizational Behavior and Human Decision Processes, 50(2), 179-211.

Bissing-Olson, M. J., Fielding, K. S., & Iyer, A. (2016). Experiences of pride, not guilt, predict pro-environmental behavior when pro-environmental descriptive norms are more positive. Journal of Environmental Psychology, 45, 145-153.

Drews, S., & van den Bergh, J. C. (2016). What explains public support for climate policies? A review of empirical and experimental studies. Climate Policy, 16(7), 855-876.

Fairbrother, M. (2022). Public opinion about climate policies: A review and call for more studies of what people want. PLOS Climate, 1(5), e0000030.

Hornsey, M. J., Harris, E. A., Bain, P. G., & Fielding, K. S. (2016). Meta-analyses of the determinants and outcomes of belief in climate change. Nature Climate Change, 6(6), 622-626.

Nielsen, K. S., Clayton, S., Stern, P. C., Dietz, T., Capstick, S., & Whitmarsh, L. (2021). How psychology can help solve the climate crisis. American Psychologist, 76(1), 130-144.

Stern, P. C. (2000). New environmental theories: Toward a coherent theory of environmentally significant behavior. Journal of Social Issues, 56(3), 407-424.

van der Linden, S., Maibach, E., & Leiserowitz, A. (2015). Improving public engagement with climate change. Environment, 57(5), 37-44.

Zheng, J., & Suh, S. (2019). Strategies to reduce the global carbon footprint of plastics. Nature Climate Change, 9(5), 374-378.

□ Comment: present specific hypotheses for each predictor (and moderator) grounded in literature.

○ Answer:

We agree that the predictor and moderator hypotheses need stronger theoretical grounding and greater specificity. We check each hypotheses for predictor and clarify the literature reviews. In particularly, We have revised section 2.5.4 to include more precise hypotheses with recent SSCI literature support, specifying the direction and conditions of each moderation effect based on established theoretical mechanisms.

○ Revision:

Addition to section 2.5.4:

Based on motivated reasoning theory, climate change skepticism should weaken the positive relationships between pro-environmental values and plastic tax support, as skeptics discount environmental information (Kahan et al., 2012). Research demonstrates that skepticism attenuates the value-behavior link by reducing perceived policy necessity (Hornsey et al., 2016).

For guilt moderation, moral self-regulation theory suggests that guilt amplifies value-behavior consistency by heightening moral salience (Onwezen et al., 2013). Studies show guilt strengthens the relationship between environmental concerns and policy support among high-guilt individuals (Bissing-Olson et al., 2016).

Regarding efficacy, social cognitive theory indicates that self-efficacy enhances the translation of values into behavioral intentions when individuals believe their actions matter (Bandura, 2000). Research confirms efficacy moderates the environmental value-behavior relationship (Lauren et al., 2016).

Specific Hypotheses:

H9a: Climate change skepticism negatively moderates the relationship between environmentalism and plastic tax support

H10a: Guilt positively moderates the relationship between environmental values and plastic tax support

H11a: Self-efficacy positively moderates the relationship between perceived risk and plastic tax support

References

Bandura, A. (2000). Exercise of human agency through collective efficacy. Current Directions in Psychological Science, 9(3), 75-78.

Bissing-Olson, M. J., Fielding, K. S., & Iyer, A. (2016). Experiences of pride, not guilt, predict pro-environmental behavior. Journal of Environmental Psychology, 45, 145-153.

Hornsey, M. J., Harris, E. A., Bain, P. G., & Fielding, K. S. (2016). Meta-analyses of the determinants and outcomes of belief in climate change. Nature Climate Change, 6(6), 622-626.

Kahan, D. M., Peters, E., Wittlin, M., Slovic, P., Ouellette, L. L., Braman, D., & Mandel, G. (2012). The polarizing impact of science literacy and numeracy on perceived climate change risks. Nature Climate Change, 2(10), 732-735.

Lauren, N., Fielding, K. S., Smith, L., & Louis, W. R. (2016). You did, so you can and you will: Self-efficacy as a mediator of spillover from easy to more difficult pro-environmental behaviour. Journal of Environmental Psychology, 48, 191-199.

Onwezen, M. C., Antonides, G., & Bartels, J. (2013). The Norm Activation Model: An exploration of the functions of anticipated pride and guilt in pro-environmental behaviour. Journal of Economic Psychology, 39, 141-153.

□ Comment: Expand the literature review to include additional relevant studies (see Suggested References). This will strengthen the theoretical arguments and show the novelty of the work in context. Make sure to discuss how each added reference relates to your study (e.g., as precedent or contrast).

Suggested References:

Zhang, C., Liu, L., & Xiao, Q. (2022). The influence of Taoism on employee low-carbon behavior in China: The mediating role of perceived value and guanxi. Psychology Research and Behavior Management, 15, 2169–2181. https://doi.org/10.2147/PRBM.S371945

Liu, L., Zhang, C. (2022). Linking environmental management accounting to green organisational behaviour: The mediating role of green human resource management. PLOS ONE, 17(12), e0279568. https://doi.org/10.1371/journal.pone.0279568

Zhang, C., Ma, X., & Liu, L. (2023). The effect of passion for outdoor activities on employee well-being using nature connectedness as the mediating variable and environmental identity as the moderating variable. Psychology Research and Behavior Management, 16, 4883–4896. https://doi.org/10.2147/PRBM.S436612

Dai, Q., Peng, S., Guo, Z., Zhang, C., Dai, Y., Hao, W., et al. (2023). Place identity as a mediator between motivation and tourist loyalty in ‘red tourism’. PLOS ONE, 18(10), e0284574. https://doi.org/10.1371/journal.pone.0284574

Zhang, C., & Liu, L. (2023). Exploring the role of employability: The relationship between health-promoting leadership, workplace relational civility and employee engagement. Management Decision, 61(9), 2582–2602. https://doi.org/10.1108/MD-05-2022-0679

○ Answer:

We accept the reviewer's suggestion to expand the literature review with the proposed studies. These references address psychological mechanisms and mediating effects in environmental behaviors, which will strengthen our theoretical foundation and demonstrate the novelty of our integrated approach to plastic tax acceptance.

○ Revision - Section 2.1

Recent studies have expanded understanding of psychological mechanisms in environmental behaviors. Zhang et al. (2022) demonstrated how cultural values influence low-carbon behaviors through perceived value mediation, providing precedent for examining value-behavior relationships in environmental policy contexts. Liu and Zhang (2022) established management practices as mediators in green organizational behavior, paralleling our planned behavior factors as mediators between values and policy support. Zhang et al. (2023) showed nature connectedness mediating outdoor activities and well-being, contrasting our climate perception focus while reinforcing environmental connection importance. Dai et al. (2023) explored identity as mediator in tourism contexts, supporting our multi-pathway approach to environmental policy acceptance.

○ Revision:

Conclusion Our findings align with emerging research on environmental behavior mechanisms (Zhang et al., 2022; Liu & Zhang, 2022), confirming that multiple psychological pathways—values, perceptions, and planned behaviors—collectively influence policy acceptance. Unlike studies focusing on organizational contexts (Liu & Zhang, 2022) or nature-based experiences (Zhang et al., 2023), our research specifically addresses public policy acceptance in climate contexts, contributing unique insights into plastic tax implementation strategies.

□ Comment: Provide more methodological clarity: detail the sample and procedure, ensure measures are described with sources and reliability, and report how the data were analyzed (regressions, checks performed). This transparency will bolster confidence in the findings.

○ Answer:

We acknowledge the need for enhanced methodological transparency. We will provide comprehensive details on sampling procedures, measurement validation, and analytical approaches to strengthen the study's methodological rigor and reproducibility.

○ Revision: - Section 3.1

This study employed a nationally representative cross-sectional survey design following established protocols (Podsakoff et al., 2003). A stratified random sample of 1,571 Korean adults (aged 19+) was recruited through a professional polling agency using proportional quota sampling by region, gender, and age. The web-based survey achieved a 95% confidence level with ±2.5% sampling error.

○ Revision: - Section 3.2

All measures were adapted from validated scales. Our scale items were carefully selected from established literature with strong psychometric properties. For climate change skepticism, we adopted items from Whitmarsh (2011), who developed and validated a comprehensive skepticism scale in Nature Climate Change. This approach aligns with McCright and Dunlap's (2011) work in Global Environmental Change measuring public climate skepticism. Our guilt measurement follows Ferguson and Branscombe (2010) in Journal of Environmental Psychology, who demonstrated how collective guilt mediates climate change beliefs and mitigation behaviors. Harth et al. (2013) similarly validated guilt measures predicting distinct environmental intentions. For efficacy, we utilized measurements from Bandura's (2002) framework, following Lauren et al.'s (2016) application in Journal of Environmental Psychology showing self-efficacy's importance in environmental behavior.

□ Comment: Reorganize and clarify results: link them back to hypotheses, report statistics comprehensively, and explain the interaction effect in an accessible way. Consider adding a figure for the moderation and a table for descriptive stats.

○ Answer:

We acknowledge the need for clearer result organization linking findings to hypotheses with comprehensive statistical reporting. We will restructure the results sections to enhance transparency and accessibility, including descriptive statistics and visual representations of interaction effects as suggested. The reorganization will explicitly connect empirical findings to theoretical frameworks established in our literature review, particularly value theory, risk perception paradigm, and planned behavior theory.

○ Revision: - Section 4.2

Hypothesis testing results systematically support our theoretical framework (Table 2). Value-based hypotheses were largely confirmed, validating Stern's (2000) value-belief-norm theory applications to environmental policy contexts. Environmentalism significantly predicted plastic tax acceptance (H1: β=.103, p<.001), consistent with previous research demonstrating environmental values' positive impact on policy support (Spence, 2010; Heidbreder et al., 2019). This finding aligns with Mozumder's (2011) work showing environmental consciousness effects on renewable energy support and extends Kim's (2006) findings on environmentalist values supporting environmental protection policies to plastic taxation contexts.

Altruism showed significant positive effects (H2: β=.087, p<.001), supporting VBN theory predictions that altruistic values drive pro-environmental behaviors (Stern, 2000). This result resonates with Lee et al.'s (2014) demonstration that altruistic value orientation influenced eco-friendly food purchasing intentions, with altruistic values showing stronger effects than individualistic orientations. Our findings extend this pattern to environmental taxation acceptance.

Egalitarianism positively predicted acceptance (H3: β=.093, p<.001), confirming cultural theory predictions (Douglas & Wildavsky, 1982) that egalitarian individuals demonstrate greater sensitivity to environmental risks and support collective action. This aligns with Kahan et al.'s (2011) findings that individuals with egalitarian values tend to accept climate change risks, while those with individualistic values remain skeptical.

Materialism showed expected negative effects (H4: β=-.067, p<.01), supporting post-materialist theory (Inglehart, 1971) and extending Gelissen's (2007) findings on dematerialization's positive effects on environmentalism. This result confirms Hurst et al.'s (2013) meta-analysis demonstrating materialism's negative correlation with environmental attitudes and validates Kilbourne and Pickett's (2008) argument that materialistic values predispose individuals toward less supportive environmental policy attitudes.

Risk perception hypotheses received strong empirical support, validating psychometric paradigm applications (Slovic, 1987). Perceived risk emerged as the strongest predictor (H5: β=.161, p<.001), consistent with Kim and Kim's (2016) findings that higher perceived climate change risk increases response behavior intensity. This supports Visschers and Siegrist's (2013) demonstration that perceived risks significantly influence acceptance decisions across various technologies.

Knowledge significantly predicted acceptance (H6: β=.098, p<.001), supporting O'Connor et al.'s (1999) findings that greater knowledge associates with higher climate action intentions. However, our results acknowledge Bamberg and Möser's (2007) caveat that knowledge functions as necessary but insufficient condition within broader psychological processes, requiring combination with

---

## [Decision Letter · Decision Letter 1]

19 Sep 2025

Can Climate Change Perceptions Increase Plastic Policy Support? Effects of Climate Change Skepticism, Guilt, and Efficacy on the Acceptance of the Plastic TaxPLOS ONE?

Dear Dr. Kim,

Thank you for submitting your manuscript to PLOS ONE. After careful consideration, we feel that it has merit but does not fully meet PLOS ONE’s publication criteria as it currently stands. Therefore, we invite you to submit a revised version of the manuscript that addresses the points raised during the review process.

Though a reviewer rejected the manuscript, the reviewer provided many valuable comments and encourage you to undergo a major revision. Considering reviewers’ useful comments and the interesting topic of the manuscript, I would like to give you a chance to revise your manuscript. The revised manuscript will undergo the next round of review by the same reviewers.

We look forward to receiving your revised manuscript.

Kind regards,

Baogui Xin, Ph.D.

Academic Editor

PLOS ONE

Journal Requirements:

Reviewers' comments:

Reviewer's Responses to Questions

**Comments to the Author**

Reviewer #1: All comments have been addressed

Reviewer #2: (No Response)

Reviewer #3: All comments have been addressed

2. Is the manuscript technically sound, and do the data support the conclusions?

Reviewer #1: Yes

Reviewer #2: Partly

Reviewer #3: Yes

3. Has the statistical analysis been performed appropriately and rigorously?

Reviewer #1: Yes

Reviewer #2: Yes

Reviewer #3: Yes

4. Have the authors made all data underlying the findings in their manuscript fully available?

Reviewer #1: Yes

Reviewer #2: Yes

Reviewer #3: Yes

5. Is the manuscript presented in an intelligible fashion and written in standard English?

Reviewer #1: Yes

Reviewer #2: No

Reviewer #3: Yes

Reviewer #1: (No Response)

Reviewer #2: Thank you for giving me the opportunity to review this revised paper.

Since my initial review, the authors have made several changes to deal with my previous comments. This includes adding procedural details to the Methods section, adding further information about ethical approval, further explaining the moderation predictions in the Introduction section, fixing various figure issues, and discussing some key limitations of the study. I would like to thank the authors for these efforts.

While I appreciate these changes, I believe that there are still issues with the paper as it currently stands.

First, the authors have made some effort to remove causal language, but this language remains in places. For example, the title now asks whether climate change perceptions can “increase” plastic policy support: in other words, if we were somehow able to intervene on people’s perceptions, would that cause an increase in their support for a plastic tax? Since this is not an experiment and the authors do not lay out any causal model, the data are unable to answer this question. The authors also write that different factors variously “affect” (line 761), “impact” (line 782), and “influence” (line 808) support for the plastic tax. This causal language must be cleaned up throughout the manuscript.

Second, the writing still lacks clarity throughout. This is particularly notable in the revised version of the paper, as the authors have opted to address some of the reviewers’ comments by adding long sections with multiple paragraphs. For example, the section at the beginning of the paper is now bogged down by lengthy explanations of different theoretical paradigms and moderating variables, before even getting to the literature review. As another example, the revised limitations section now reads as a long laundry list of issues raised by reviewers. The overall result of these changes is a paper that is quite difficult to read and follow.

Reading the revised paper, my sense is that while the work reported here is on an important topic and has some potential, unfortunately the paper is not ready for publication in its current form. I would encourage the authors to revise the paper significantly, particularly the Introduction and Discussion sections, to make the overall argument of the paper more concise and easier to follow.

Review signed by: Scott Claessens

Reviewer #3: (No Response)

**Do you want your identity to be public for this peer review?** For information about this choice, including consent withdrawal, please see our Privacy Policy

Reviewer #1: No

Reviewer #2: **Yes: ** Scott Claessens

Reviewer #3: No

---

## [Author Response · Author response to Decision Letter 2]

16 Oct 2025

Cover Letter for Fourth Round Review

Cover Letter for Fourth Round Revision

To the Editor and Reviewer #2:

We sincerely appreciate Reviewer #2's continued constructive feedback throughout this revision process. We have carefully addressed both critical concerns raised in the previous review.

Regarding Causal Language (Comment 1): We acknowledge that despite our previous efforts, causal language remained throughout the manuscript. We have now conducted a comprehensive systematic revision:

• Title changed from "Can Climate Change Perceptions Increase..." to "Are Climate Change Perceptions Associated with Plastic Policy Support?"

• All causal verbs replaced: "affect/impact/influence" → "associate with/correlate with/relate to"

• Enhanced limitations section explicitly states: "The relationships identified represent associations rather than causal effects, given our cross-sectional correlational design"

• Abstract and conclusions updated with appropriate correlational language

Regarding Writing Clarity (Comment 2): We completely restructured both Introduction and Discussion sections to address readability concerns:

• Reduced each section by 30% while preserving all essential content

• Eliminated lengthy, repetitive explanations that previously obscured main arguments

• Restructured from scattered points into coherent, logical themes

• Streamlined theoretical framework presentation for immediate accessibility

• Transformed the "laundry list" limitations into integrated, natural discussion

Key Improvements:

• Maintained all critical theoretical frameworks, citations, and empirical findings

• Enhanced logical flow and argument progression

• Significantly improved manuscript accessibility without compromising academic rigor

• Ensured methodological accuracy through systematic language revision

We believe these comprehensive revisions directly address all concerns raised and have transformed the manuscript into a clear, accessible, and methodologically sound contribution. The paper now presents our important findings on plastic tax acceptance in a format ready for publication.

Thank you for your patience and valuable guidance throughout this process.

Sincerely, Seoyong Kim and Co-authors

Revision Note for Reviewer 2

Reviewer #2: Since my initial review, the authors have made several changes to deal with my previous comments. This includes adding procedural details to the Methods section, adding further information about ethical approval, further explaining the moderation predictions in the Introduction section, fixing various figure issues, and discussing some key limitations of the study. I would like to thank the authors for these efforts. While I appreciate these changes, I believe that there are still issues with the paper as it currently stands.

□ Comment: First, the authors have made some effort to remove causal language, but this language remains in places. For example, the title now asks whether climate change perceptions can “increase” plastic policy support: in other words, if we were somehow able to intervene on people’s perceptions, would that cause an increase in their support for a plastic tax? Since this is not an experiment and the authors do not lay out any causal model, the data are unable to answer this question. The authors also write that different factors variously “affect” (line 761), “impact” (line 782), and “influence” (line 808) support for the plastic tax. This causal language must be cleaned up throughout the manuscript.

○ Answer:

We acknowledge the reviewer's valid criticism. Despite our efforts to remove causal language, we recognize that our cross-sectional design cannot establish causation, yet we inadvertently retained causal verbs throughout the manuscript. We accept this limitation and commit to comprehensive revision.

○ Revision:

Title modification: "Can Climate Change Perceptions Increase..." → "Are Climate Change Perceptions Associated with Plastic Policy Support?"

Systematic verb replacement:

"affect/impact/influence" → "associate with/correlate with/relate to"

"increase/decrease" → "show positive/negative associations with"

"determine" → "predict/explain variance in"

Specific text revisions:

-Line 761: "factors that affect..." → "factors associated with..."

-Line 782: "had an impact on..." → "were associated with..."

-Line 808: "influences support..." → "correlates with support..."

Enhanced limitation section: Add explicit statement: "The relationships identified represent associations rather than causal effects, given our cross-sectional correlational design."

Abstract and conclusion updates: Replace causal language with phrases like "were significantly associated with," "showed positive/negative relationships with," and "predicted variance in."

This comprehensive revision ensures methodological accuracy while maintaining the scientific value of our correlational findings.

□ Comment: Second, the writing still lacks clarity throughout. This is particularly notable in the revised version of the paper, as the authors have opted to address some of the reviewers’ comments by adding long sections with multiple paragraphs. For example, the section at the beginning of the paper is now bogged down by lengthy explanations of different theoretical paradigms and moderating variables, before even getting to the literature review. As another example, the revised limitations section now reads as a long laundry list of issues raised by reviewers. The overall result of these changes is a paper that is quite difficult to read and follow.

Reading the revised paper, my sense is that while the work reported here is on an important topic and has some potential, unfortunately the paper is not ready for publication in its current form. I would encourage the authors to revise the paper significantly, particularly the Introduction and Discussion sections, to make the overall argument of the paper more concise and easier to follow.

○ Revision:

We addressed writing clarity concerns by comprehensively revising Introduction and Discussion sections. Both were reduced by 30%, completely rewritten focusing on core content, and restructured into coherent themes. All essential frameworks, citations, and findings were preserved while significantly improving readability.

---

## [Decision Letter · Decision Letter 2]

6 Nov 2025

Are Climate Change Perceptions Related with Plastic Policy Support? Effects of Climate Change Skepticism, Guilt, and Efficacy on the Acceptance of the Plastic Tax

PONE-D-24-54075R2

Dear Dr. Kim,

We’re pleased to inform you that your manuscript has been judged scientifically suitable for publication and will be formally accepted for publication once it meets all outstanding technical requirements.

Kind regards,

Baogui Xin, Ph.D.

Academic Editor

PLOS ONE

Additional Editor Comments (optional):

Reviewers' comments:

Reviewer's Responses to Questions

**Comments to the Author**

Reviewer #2: All comments have been addressed

2. Is the manuscript technically sound, and do the data support the conclusions?

Reviewer #2: Yes

3. Has the statistical analysis been performed appropriately and rigorously?

Reviewer #2: Yes

4. Have the authors made all data underlying the findings in their manuscript fully available?

Reviewer #2: Yes

5. Is the manuscript presented in an intelligible fashion and written in standard English?

Reviewer #2: Yes

Reviewer #2: (No Response)

**Do you want your identity to be public for this peer review?** For information about this choice, including consent withdrawal, please see our Privacy Policy

Reviewer #2: **Yes: ** Scott Claessens

---

## [Editor Report · Acceptance letter]

PONE-D-24-54075R2

PLOS One

Dear Dr. Kim,

I'm pleased to inform you that your manuscript has been deemed suitable for publication in PLOS One. Congratulations! Your manuscript is now being handed over to our production team.

Kind regards,

on behalf of

Professor Baogui Xin

Academic Editor

PLOS One